# Environmental DNA metabarcoding of water samples as a tool for monitoring Iberian freshwater fish composition

**Andrea Corral-Lou**[1,2], **Ignacio Doadrio**[1]*

**1** Biodiversity and Evolutionary Biology Department, Museo Nacional de Ciencias Naturales, CSIC, José Gutiérrez Abascal, Madrid, Spain, **2** Consultores en Biología de la Conservación S.L., Daoiz, Madrid, Spain

* doadrio@mncn.csic.es

**Data Availability Statement:** All relevant data are within the paper and its Supporting information files.

**Funding:** Industrial Doctorate of Community of Madrid. Grant/Award Number: IND2017/AMB-

## Abstract

Environmental DNA (eDNA) metabarcoding has been increasingly used to monitor the community assemblages of a wide variety of organisms. Here, we test the efficacy of eDNA metabarcoding to assess the composition of Iberian freshwater fishes, one of the most endangered groups of vertebrates in Spain. For this purpose, we sampled 12 sampling sites throughout one of Spain's largest basins, the Duero, which is home to approximately 70% of the genera and 30% of the primary freshwater fish in Spain. We sampled these sampling sites in the summer by using electrofishing, a traditional sampling method, and eDNA metabarcoding of river water samples using the mitochondrial 12S rRNA gene (*12S*) as a marker. We also resampled four of these sampling sites in autumn by eDNA. We compared the results obtained through eDNA metabarcoding with those of electrofishing surveys (ones conducted for the present study and past ones) and assessed the suitability of *12S* as an eDNA metabarcoding marker for this group of freshwater fishes. We found that the *12S* fragment, analysed for 25 Iberian species, showed sufficient taxonomic resolution to be useful for eDNA approaches, and even showed population-level differences in the studied populations across the tissue samples for *Achondrostoma arcasii*. In most cases, a greater number of species was detected through eDNA metabarcoding than through electrofishing. Based on our results, eDNA metabarcoding is a powerful tool to study the freshwater fish composition in the Iberian Peninsula and to unmask cryptic diversity. However, we highlight the need to generate a local genetic database for *12S* gene for such studies and to interpret the results with caution when studying only mitochondrial DNA. Finally, our survey shows that the high detection sensitivity of eDNA metabarcoding and the non-invasiveness of this method allows it to act as a detection system for species of low abundance, such as early invasive species or species in population decline, two key aspects of conservation management of Spanish freshwater fishes.

## Introduction

Approximately 9.5% of the planet's animal biodiversity is found in freshwater ecosystems, with vertebrates representing one-third of this percentage [1, 2]. Of the freshwater vertebrates,

7699; Duero Hydrographic Confederation. Grant/
Award Number: Life13 nat/es/000772.

**Competing interests:** The authors have declared
that no competing interests exist.

fishes represent one of the most abundant groups, contributing to a large part of the biodiversity found in rivers [3]. The monitoring of this group is, therefore, fundamental for ecosystem conservation and the sustainability of biological resources [4–6]. Freshwater fish monitoring has traditionally been conducted by using direct capture approaches such as electrofishing, netting and trapping [7]. Sometimes, these traditional methods are hampered by the geomorphological characteristics of a river, net location in the water column or habitat, mesh size of nets, unfavorable weather and water conditions and study design (e.g., the need for legal permits to capture specimens or for experts to make the morphological identifications). They are also invasive techniques: in most cases, the species of interest must be handled or captured to obtain a positive identification, which can compromise the viability of the species, especially if they are endangered [8]. Traditional (capture) methods also tend to be inefficient when dealing with the detection of species present at a low density, such as those suffering a population decline or recently introduced species. Their exclusive use may result in an underestimation of freshwater fish diversity or necessitate extremely laborious sampling efforts.

Over the last decade, an alternative method for detecting the presence of freshwater fishes has been developed based on the use of environmental DNA (eDNA), i.e., extra-organismal genetic material suspended in environmental samples such as water or sediment [9]. In recent years, eDNA approaches utilizing water samples have been broadly used to monitor the biodiversity of fish communities or specific species in freshwater ecosystems [10–17]. This technique has proven to be, among other factors, more time and cost effective than traditional methods, and has shown greater sensitivity in detecting low-abundance species [18–20]. Moreover, the difficulties associated with directly sampling fishes (e.g., river conditions or depth) are overcome using eDNA as only a water sample is needed. However, eDNA approaches have limitations related to making inferences across space and time, the presence versus viability of populations, the PCR artifacts, or the ability of eDNA as a quantitative tool [20, 21]. This technique also faces challenges associated with low-quality, low-quantity of DNA in the samples subject to other factors such as capture and extraction efficacy, the presence of inhibitors and contamination challenges which can be lead to presence of false positives or negatives [20, 21]. Therefore, even if it has been demonstrated that this technique is highly sensitive to the detection of low DNA concentrations, appropriate critical experimental design considerations are essential.

One of the most important points to consider when using eDNA metabarcoding for freshwater fishes is the choice of the genetic marker used to amplify target taxa [22]. The use of mitochondrial markers for eDNA metabarcoding has been shown to be successful for some groups, and more advantageous compared with nuclear ones, likely due to the higher copy number per cell and less susceptibility to environmental degradation of mitochondrial DNA (mtDNA) [18, 23]. In particular, mitochondrial 12S rRNA (*12S*) has been the most widely used gene marker for eDNA metabarcoding of bony fishes [24]. The strength of *12S* markers for eDNA metabarcoding lies primarily in its more fish-specific amplification than other mitochondrial markers such as cytochrome c oxidase I (*COI*) and cytochrome b (*MT-CYB*) [22, 24, 25]. Despite this, the use of this fragment for eDNA metabarcoding of freshwater fishes has some drawbacks. Its variability, although moderate, is lower than *COI* and *MT-CYB*, and sometimes, it shows a low level of taxonomic resolution and cannot be used to distinguish among some fish taxa [15, 26–28]. Moreover, the reference database for *12S* is not as extensive as those for *MT-CYTB* or *COI* markers in public repositories such as GenBank, widely used in phylogenetic and phylogeographic studies, making the taxonomic assignment of freshwater fishes more difficult [22].

Primary freshwater fishes (i.e., whose ancestors entered inland waters much earlier, cannot survive in seawater and are thus strictly confined to fresh water) [29–31] of the Iberian

Peninsula have been widely studied in a phylogenetic and phylogeographic context [e.g., 32–35]. All phylogenetic and phylogeographic studies in Iberian primary freshwater fishes have been conducted using the *MT-CYTB* and *COI* markers, while no surveys have been carried out with the *12S* marker [36–41]. In fact, unlike for the other markers, there is no *12S* database for native Iberian freshwater fishes.

Compared with other vertebrate groups in the Iberian Peninsula, the freshwater fishes in this area consist of a greater proportion of endemic species and represent one of the most endemic fish faunas in the Mediterranean region [42]. Specifically, of the 68 species present in the Iberian Peninsula, 55 are continental and 44 are endemic, and some of those that are not considered endemic have their range limited to a small portion of southern France, such as *Phoxinus bigerri* Rafinesque, 1820 and *Barbatula hispanica* (Lelek, 1987) [42, 43]. The Duero Basin, with a surface area of 97,290 km$^2$, is the largest basin in the Iberian Peninsula (81% of it is in Spain, and 19% is in Portugal). The Duero Basin currently hosts 70% of the genera and 30% of Spain's native freshwater fishes approximately, considering *Gobio lozanoi* Doadrio & Madeira, 2004 and *Tinca tinca* (Linnaeus, 1758), whose origin in the basin remains undetermined [42, 44, 45]. Of the native primary freshwater species found in the Duero Basin, 64% are included within one of the three most critical categories on the IUCN Red List (Vulnerable, Endangered and Critically Endangered; IUCN, 2020), including one of the most endangered Iberian endemic fish species, *Achondrostoma salmantinum* Doadrio & Elvira 2007 with its distribution area restricted to a few rivers in the southwestern Duero Basin. The main threats of these species are associated at anthropogenic factors, which in some cases in having led to an evident decline in populations of several of the species [42]. The main anthropogenic threat is the 18 large reservoirs and 3257 small dams that occur throughout the Duero Basin (https://www.miteco.gob.es; National River restoration strategy 2022–2030). This reservoirs or dams modify the dynamics of rivers, the habitat of native species, land use (93% of the reservoir water is used for irrigation) and are the main source of introduction of invasive species for recreational use [46–50]. Specifically, the Duero Basin hosts a large variety of non-native species, currently recorded a total of 15 non-native species (https://www.chduero.es/), being the presence of non-native species, a major threat to the sustainability of native species [51]. While the actual impact of invasive species on native species is often unknown, they can affect native wildlife through a variety of factors. Invasive species can compete for niches or resources, prey on native fauna, transfer pathogens, alter habitat, or cause genetic introgression, ultimately leading to loss of genetic diversity [52–56].

Duero Basin also harbors a great intraspecific genetic diversity for multiples native Iberian freshwater fish species, as revealed by multiple population genetics surveys [32, 40, 41]. On the one hand, for several species within this basin that are also distributed in other Iberian basins, such as *Cobitis vettonica* (Doadrio & Perdices, 1997) [57], *Squalius alburnoides* (Steindachner, 1866) [58] and *S. carolitertii* (Doadrio, 1988) [59], population genetics studies have revealed that Duero Basin populations behave as an independent evolutionary lineage [40, 60, 61]. On the other hand, a strong genetic structure within the populations distributed throughout the Duero Basin has been detected in other species such as *C. calderoni* Bacescu, 1962, *Pseudochondrostoma duriense* (Coelho, 1985) or *Achondrostoma arcasii* (Steindachner, 1866) [32, 40, 62]. For all these reasons, the conservation of freshwater fish populations in the Duero helps to protect not only the species richness of the Iberian Peninsula but also intraspecific genetic richness.

Due to the large number of both native and non-native species inhabiting the Duero Basin in relation to Spain's entire freshwater fish fauna, the strong threat and the genetic richness of their native populations, accurate knowledge of the composition of this basin's ichthyofauna is of critical importance for Spanish freshwater fish conservation. The main objective of this

study is to assess eDNA metabarcoding as a tool for monitoring freshwater fish fauna from Duero Basin comparing with electrofishing results and to explore the use of *12S* as a marker for this approach. For this purpose, we sequenced a fragment of *12S* from tissue sample of all species inhabiting in the Duero Basin, tested the taxonomic resolution of this fragment and its ability to distinguish local variation and built a local *12S* reference database for around of 30% of Spanish native freshwater fishes. Finally, we provide useful information on the presence of non-native species detected through eDNA, which is considered one of the main threats for Iberian native freshwater fishes.

## Materials & methods

### Ethics statement

This investigation was conducted entirely in accordance with ethical standards and Spanish legislation. Permission for sampling was issued by the Dirección General de Medio Ambiente de la Junta de Castilla y León from Spain.

### Sampling sites

A total of 12 sampling sites were analyzed in this study. These localities were distributed throughout the entire Spanish Duero Basin (Fig 1; Table 1). The selection of the sampling sites was based on ichthyological electrofishing samplings made in 2001, 2009 and 2010 from previous projects attempting to obtain a representation of all the fishes that can be found in the Duero Basin [42, 63]. The samples for eDNA metabarcoding were collected just prior to electrofishing in all sampling sites to avoid DNA contamination from the electrofishing gear

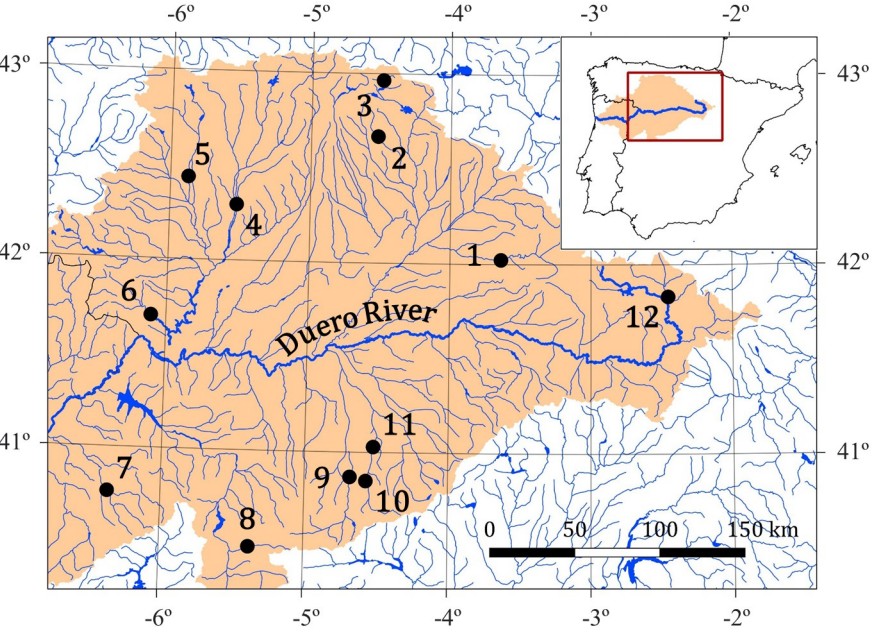

**Fig 1. Sampling sites for this study.** Numbers correspond to those used in Table 1. The orange shading delimits the Duero Basin. Maps in this figure were made using the free software Qgis v.3.10.7. Inland water and administrative areas shapefiles from Spain were downloaded from data provided by DIVA-GIS project (https://www.diva-gis.org/gdata) and the Duero Basin shapefile was downloaded from data provided by the Ministerio para la Transición Ecológica y el Reto Demográfico (https://www.miteco.gob.es/es/cartografia-y-sig/ide/descargas/agua/cuencas-y-subcuencas.aspx).

**Table 1. Sampling sites include in the present study.** For each sampling site the following has been indicated: number on the map (Nº); Collection date of the samples in summer and autumn (Date M1 and Date M2 respectively); Name of the samples collected in summer and autumn (ID M1 and ID M2 respectively); Name of the river (River); The municipality in which it is located (Locality); The size of the aquatic system (Size); River section (Section); Land use type (Land use); Degree of impact of the river (Impact); Sub-basin to which it belongs (Sub-basin).

| Nº | Date M1 | Date M2 | ID M1 | ID M2 | River | Locality | Size | River section | Land use | Impact | Sub-basin |
|----|---------|---------|-------|-------|-------|----------|------|---------------|----------|--------|-----------|
| 1 | 6/7/2020 | | AT5138 | | Arlanza | Quintanilla del Agua, Burgos | Small river | Middle | Farmland and forestland | Low | Pisuerga |
| 2 | 7/7/2020 | | AT5141 | | Boedo | Báscones de Ojeda, Palencia | Stream | Upper | Urban | Low | Pisuerga |
| 3 | 7/7/2020 | | AT5140 | | Pisuerga | San Salvador de Cantamuda, Palencia | Small river | Upper | Forestland | Low | Pisuerga |
| 4 | 10/7/2020 | 23/11/2020 | AT5144 | AT5166 | Esla | Valecia de Don Juan, León | Large river | Middle | Farmland and recreational area | Low | Esla |
| 5 | 10/7/2020 | 23/11/2020 | AT5145 | AT5167 | Órbigo | Barrio de Buenos Aires, León | Small river | Middle | Farmland, livestock and recreational area | Medium | Esla |
| 6 | 10/7/2020 | | AT5143 | | Aliste | Vegalatrave, Zamora | Small river | Lower | Farmland | Medium | Esla |
| 7 | 20/7/2020 | | AT5152 | | Yeltes | Martín de Yeltes, Salamanca | Stream | Middle | Farmland and livestock | Medium | Huebra |
| 8 | 20/7/2020 | | AT5153 | | Corneja | San Bartolomé de Corneja, Ávila | Stream | Middle | Farmland and livestock | Low | Tormes |
| 9 | 24/7/2020 | 23/11/2020 | AT5155 | AT5168 | Adaja | Blascosancho, Ávila | Small river | Upper | Forestland and recreational area | Low | Adaja |
| 10 | 24/7/2020 | | AT5156 | | Voltoya | Pinar de Puente Viejo, Ávila | Stream | Middle | Farmland and livestock | High | Adaja |
| 11 | 24/7/2020 | | AT5154 | | Voltoya | Juarros de Voltoya, Segovia | Reservoir | Upper | Farmland and reservoir | High | Adaja |
| 12 | 7/7/2020 | 25/11/2020 | AT5142 | AT5169 | Tera | Tardesillas, Soria | Small river | Middle | Farmland and recreational area | Low | Tera |

between sampling localities. In the sampling site Juarros de Voltoya (11; Table 1 and Fig 1), electrofishing was impractical within the reservoir itself, therefore we sampled the area just a few meters from the mouth of the reservoir in the Voltoya River.

## Environmental DNA sampling and extraction

Sampling was mainly carried out during July 2020 (12 sampling sites; M1; Table 1). We also resampled four of the 12 sites in October 2020, which were associated with a recreational area and therefore a greater possibility of finding alien eDNA (M2; Table 1). All the used material was first disinfected with 20% bleach. The water samples were taken at two different points in the river, trying to cover all the heterogeneity of mesohabitats. The water was collected through a sterile bottle in the parts accessible to walk from the river with the use of gloves. A sample of the vertical water column was taken by submerging the bottle horizontally to the end and lifting it to the surface. *In situ* filtration was conducted on the riverbank to avoid cross-contamination and eDNA degradation. All water samples were filtered through Nalgene ™ Reusable Filter Holders with Receivers fitted with Supor®-200 Membrane Filters (Pall Corporation, Life Sciences, Ann Arbor, MI, USA) with a pore size of either 0.45 μm or 0.2 μm. ™ Reusable Filter Holders were connected to a vacuum pump which, in turn, was connected to a generator to allow the water to pass through the filters. For each locality, 1 L of distilled water was first filtered as a negative control (blank, BL) from the laboratory and then two 1-L samples of the river water (L1 and L2) were collected and filtered. Depending on the turbidity of the water,

we had to use between one and three 0.45 μm-filters per liter. The filtered water samples were then re-filtered through a 0.2-μm filter to recover the maximum amount of eDNA present in the water. Each filter was stored directly in 1 ml of ATL buffer. Within 24 hours, 35 μl of proteinase K was added to each sample and it was digested for 15 hours at 56 ºC.

DNA was extracted using the Qiagen DNeasy® Blood and Tissue Kit (Qiagen, Inc., Valencia, CA) following the manufacturer's protocol, except for the first step. The modification was the following: After digestion for 15 hours, 1 ml of the digestion product of all the filters of the same liter was placed in a single 15 ml falcon tube, and 1 ml of AL buffer and 1 ml of ethanol were added for each milliliter of digestion product. This solution was then passed through a spin column. After this step, the manufacturer's protocol was followed.

## Traditional sampling (electrofishing)

In all sampling sites, the traditional electrofishing sampling method has also been carried out to compare the species detected by the two methods following the European regulations (EN ISO 14011:200. Water quality Sampling of fish with electrofishing). All scientific fishing permits were obtained from the relevant authorities (Environment service of council of Castilla y León). Immediately after capture and identification, all individuals were returned to the river. Accessible River sections were selected to arrive by car and that it was possible to carry out electrofishing. At each sampling site, a river section of approximately 100 meters of distance just upstream from where the eDNA samples were taken, electrofishing sampling was carried out for 30 minutes approximately. The same transect was surveyed two following times separated by an hour. Electrofishing sampling was done in a zig zag covering all the accessible mesohabitats except in the Esla River (the only large river) where it was only possible to do electrofishing on the riverbank due to its great flow. For some specimens, a fin tissue sample was taken before they were returned to the river. Fin samples were preserved in 95% ethanol and stored at 4ºC until further processing. No individuals were sacrificed.

## Inventory of the ichthyofaunal composition

An inventory of the ichthyofaunal composition was made at each sampling site. We created a list of all species observed from the current electrofishing survey and then we added the data from the list of all species recorded from previous projects led by the second author in 2001, 2009 and 2010 [42, 63]. The species record compiled for Juarros de Voltoya Reservoir was based on sightings of species within the reservoir provided by local fishermen at the moment and on data gathered in previous years for the area just downstream of the reservoir from the projects mentioned above.

With the results obtained from this study for both eDNA metabarcoding and electrofishing, we provide useful information on the composition of the ichthyofauna, especially for non-native species. In addition, to test whether the number of detected species differed according to the sampling method used (eDNA metabarcoding and electrofishing) we used non-parametric Wilcoxon tests paired by all sample localities, since not all datasets were normally distributed. The comparison was carried out for both, considering the number of species detected by electrofishing including past and present data and considering only the present data. We also tested if there were differences in the number of species detected for summer and autumn through eDNA metabarcoding in Esla, Órbigo, Adaja and Tera.

## Library preparation and sequencing

A total of 48 samples for eDNA metabarcoding was analyzed for the 12 sampling sites (including the 4 re-sampled sites). Three samples, 1 blank (distilled water) and 2 eDNA (river water),

were analyzed for each site. For the library preparation, an approximately 250-base pair (bp) fragment of *12S* was amplified using the primers MiFish_U_F (5′ `GTCGGTAAAACTCGTGCCAGC 3′`) and MiFish_U_R (5′ `CATAGTGGGGTATCTAATCCCAGTTTG 3′`) [64]. Illumina sequencing primers were attached to these primers at the 5' ends.

The summer (M1) and autumn (M2) samples were treated independently but equally as explained below. Polymerase chain reactions (PCRs) were carried out in a final volume of 25 μL containing 2.5 μL of template DNA, 0.5 μM of the primers, 12.5 μL of Supreme NZYTaq 2x Green Master Mix (NZYTech) and ultrapure water up to 25 μL. The thermocycling conditions were as follows: an initial denaturation step at 95 ºC for 5 min, followed by 35 cycles at 95 ºC for 30 s, 55 ºC for 45 s, 72 ºC for 30 s, and a final extension step at 72 ºC for 10 min. The oligonucleotide indices required for multiplexing different libraries in the same sequencing pool were attached in a second round of PCR using same thermocycling conditions except only 5 cycles were performed and the annealing temperature was 60 ºC. A negative control containing no DNA (BL M1 and BL M2) was included in every PCR round to check for contamination during the library preparation. The libraries were run on 2% agarose gels stained with GreenSafe (NZYTech) and imaged under UV light to verify the library size. Libraries were purified using Mag-Bind RXNPure Plus magnetic beads (Omega Biotek), following the manufacturer's instructions. Then, libraries and DNA extraction blanks that yielded a band in the agarose gel were pooled in equimolar amounts according to the quantification data provided by the Qubit dsDNA HS Assay (Thermo Fisher Scientific). This pool also contained a recommended amount (1 μL) of the DNA extraction blanks that did not yield a band and of the PCR blank (BL). The pool was sequenced using the Illumina MiSeq PE300 platform.

## Preparation of the reference database for *12S*

The selection of species for the *12S* reference database was based on previous knowledge of the ichthyofauna inhabiting the Duero Basin and genetic population structure studies [39–41]. The following 26 species (25 species present in Spain and *Luciobarbus rifensis* Doadrio, Casal-Lopez & Yahyaoui, 2015 from northern Morocco) were included in the database: *Achondrostoma arcasii*; *A. salmantinum*, 2007; *Alburnus alburnus* Linnaeus, 1758; *Ameiurus melas* Rafinesque, 1820; *Barbatula barbatula* (Linnaeus, 1758); *Carassius auratus* (Linnaeus, 1758); *Cobitis calderoni*; *C. paludica* (de Buen, 1930); *C. vettonica*, 1997; *Cyprinus carpio* (Linnaeus, 1758); *Esox lucius* Linnaeus, 1758; *Gambusia holbrooki* Girard, 1859; *G. lozanoi*; *Leuciscus aspius* (Linnaeus, 1758); *Lepomis gibbosus* Linnaeus 1758; *Luciobarbus bocagei* (Steindachner, 1864); *L. rifensis* Doadrio, Casal-Lopez & Yahyaoui, 2015; *Micropterus salmoides* Lacépède, 1802; *Oncorhynchus mykiss* (Walbaum, 1792); *P. bigerri*; *Pseudochondrostoma duriense*; *P. polylepis* (Steindachner, 1864)*; Salmo trutta* Linnaeus, 1758; *Squalius alburnoides*; *S. carolitertti* and *T. tinca*. First, we downloaded all available *12S* sequences of the selected species from GenBank, which totaled 149. To these, we added the *12S* sequences of 78 specimens belonging to 19 species whose tissue had been previously sampled and stored in ethanol in the DNA and Tissue Collection at the National Museum of Natural Sciences of Madrid (MNCN–CSIC) (S1 Table in S1 File). Details of the number of specimens used for each species and the GenBank entrance number for each specimen are presented in S1 Table in S1 File. When possible, the specimens selected for sequencing came from rivers of Duero Basin.

DNA of these specimens was extracted using the Qiagen DNeasy® Blood and Tissue Kit (Qiagen, Inc., Valencia, CA), following the manufacturer's protocol. PCR was used to amplify between 217 and 229 bp of mitochondrial *12S* using the same MiFish primers used to make the libraries [64]. All reactions consisted of a final volume of 12.5 μl containing 6.25 μl of 2X

DreamTaq Green PCR Master Mix (Thermo Scientific), 0.25 μl of each primer (10 μM) and 1 μl of template DNA (final concentration 80–200 ng/μl). The following thermocycling conditions were used: initial denaturation at 95 ℃ (5 min) followed by 35 cycles of denaturation at 95 ℃ (20 s), annealing at 65 ℃ (15 s) and extension at 72 ℃ (15 s); and a final extension step at 72 ℃ (7 min). PCR products were purified with ExoSAP-IT (USB Cleveland, OH, USA), and then both strands were sequenced on a 3730xl DNA Analyzer by Macrogen Europe Inc. (http://www.macrogen.com).

All new electropherograms from the new sequences (78 sequences from tissue samples) were reviewed and cleaned one by one. Then all sequences (including GenBank sequences) were aligned using MAFFT [65], as implemented in Geneious 10.1.3 (http://www.geneious.com) [66] and collapsed into haplotypes. Finally, to cover a wider spectrum of species, all our sequences were added to an already existing reference database which contains a trained reference sets that can be used to taxonomically assign fish *12S* mitochondrial gene sequences [67, 68].

## Genetic distance

The taxonomic resolution inter-and intraspecific was examined through the genetic differentiation of the different sequences from both tissue and GenBank samples. This parameter was calculated to calibrate the parameters in the bioinformatics pipeline. Genetic differentiation for *12S* between and within species was based on the percentage of similarity calculated in Geneious 10.1.3. This parameter was calculated for the collapsed matrix of 46 haplotypes of the total of 227 sequences (149 from GenBank and 78 from the present study) for the 26 studied species. The species *A. arcasii* was divided into three independent groups based on the high level of genetic differentiation found for its populations in the Duero Basin. They were named as *A. arcasii* (1), *A. arcasii* (2) and *A. arcasii* (3).

## Metabarcoding pipeline

The overall quality of the raw sequencing reads was visualized using FastQC [69]. To merge forward and reverse sequences, we used the program FLASH2 v2.2.00 [70] and the parameter *-m* 30 *-M* 150. The primer sequences were trimmed using CUTADAPT version 1.2.1 [71], allowing for a mismatch of up to 3 bp per primer. Reads without primer sequences were removed. CUTADAPT was also used to eliminate sequences with sizes <150 bp or >200 bp since the mean size of the *12S* gene is around 170 bp [64]. We used the *multiple_split_libraries_fastq.py* script from QIIME [72] to remove low quality reads (score <20), those with 3 consecutive bp of low quality and those with less than 85% of consecutive bp of high quality. We also used this script for demultiplexing the libraries. Retained sequences were clustered into operational taxonomic units (OTUs) and chimeras were filtered out by using VSEARCH [73]. The taxonomic assignment was performed using the *assign_taxonomy.py* script from QIIME against our reference database. We performed two assignments, one with a minimum percentage of similarity threshold of 99% at species level and the other, of 97% at genera level. Subsequently, unassigned OTUs and those with less than 10 sequences within the same sample were removed. We combined both matrices and keeping the taxonomic assignment provided to 99% (at species level) for those OTUs that were assigned by both 99% and 97%. The OTUs that were only assigned to 97% were kept at the genera level. Unassigned OTUs sequences were aligned, using the Blastntool, to sequences in the GenBank nucleotide database from the National Center for Biotechnology Information (NCBI). In addition, to ensure the taxonomic assignment of the unassigned reads, we only kept reads whose cover of length of the alignment with a sequence from GenBank was 100% and had a percentage of similarity greater than 97%.

## Results

### Genetic differentiation

We analyzed 227 sequences of *12S* (149 from GenBank and 78 new sequences of tissue samples from DNA and Tissue Collection at the National Museum of Natural Sciences of Madrid) of 26 freshwater fish species (accession numbers of 78 new sequences: OP738998–OP739075) (S1 Table in S1 File). These sequences represent a total of 46 different haplotypes. The percentage of intraspecific similarity ranged from 98.3% to 100% for the native species, and from 99.7% to 100% for the non-native species (Table 2; S2 Table in S1 File). Likewise, the percentage of similarity between species varied, but it was never higher than 99%, except for the species pair *P. duriense* and *P. polylepis* (99.4%). Among the three groups of *A. arcasii*, similarity ranged from 94.3% to 97.1%. Species groups showing high interspecific similarity (97–99%) were the following: (*A. salmantinum*, *A. arcasii*, *P. duriense* and *P. polylepis)*; (*S. alburnoides* and *S. carolitertii)*; (*C. auratus* and *C. carpio*); (*C. paludica*, *C. vettonica* and *C. calderoni*); (*L. bocagei* and *L. rifensis*) and (*A. alburnus* and *L. aspius*) (Table 2 and S2 Table in S1 File).

**Table 2. Summary of the percentage of similarity detected for the *12S* gene within each species.** Species pairs/groups showing an interspecific percentage of similarity >97% are also indicated. Non-native species are marked with an asterisk. Only the percentages of interspecific similarity between the species with values between 97–99% are shown. More detailed information on percentage of similarity between the different species is provided in S2 Table in S1 File.

| Species | Intraspecific | Interspecific |
|---|---|---|
| *S. alburnoides* | 98.9–100% | 98.3% |
| *S. carolitertii* | 100% | |
| *A. arcasii* (2) | 100% | 94–98% |
| *A. arcasii* (1) | 100% | 99.4% (between *P. duriense* and *P. polylepis* species) |
| *A. salmantinum* | 99.4–100% | |
| *P. duriense* | 100% | |
| *P. polylepis* | 100% | |
| *A. arcasii* (3) | 100% | |
| *A. alburnus** | 100% | 97.7% |
| *L. aspius** | 100% | |
| *P. bigerri** | 100% | |
| *G. lozanoi* | 100% | |
| *T. tinca* | 100% | |
| *C. auratus** | 98.3–100% | 96.5–98.3% |
| *C. carpio** | 97.7–100% | |
| *L. bocagei* | 100% | 98.3% |
| *L. rifensis** | 100% | |
| *B. barbatula* | 99.4–100% | |
| *C. calderoni* | 99.4–100% | 96.6–98.3% |
| *C. paludica* | 100% | |
| *C. vettonica* | 98.8–100% | |
| *A. melas** | 100% | |
| *L. gibbosus** | 99–100% | |
| *M. salmoides** | 98.2–100% | |
| *E. lucius** | 100% | |
| *O. mykiss** | 97.7–100% | |
| *S. trutta** | 99.4–100% | |
| *G. holbrooki** | 100% | |

## Metabarcoding pipeline

Excluding the blank samples, a mean of 39,509 and 46,086 raw reads for each sampling site was obtained for M1 and M2, respectively. Samples with the lowest number of reads were L1 and L2 of Voltoya River (AT5156; M1); those with the highest values were L1 from Arlanza River (AT5138; M1) and L2 from Tera River (AT5169; M2) (Fig 2). After merging and filtration steps, 47% (sd±16%) of the raw reads comprised the final data set; of these, a mean of 85% (sd±17%) of reads for each sampling site were assigned to fish species present in our database (Fig 2; S3, S4 Tables in S1 File). After the reads were filtered for quality and taxonomically assigned, only the blanks collected from the Arlanza and Tera rivers during summer had reads assigned to fishes (Fig 2). For these two sites, the number of the final assigned reads were 31,321 and 16,526, respectively. In both cases, the final all these reads were associated with a single species or genus: *Phoxinus* for Arlanza and *Salmo* and *S. trutta* for Tera. These species were recently detected, or have been detected in the past, in the corresponding river by electrofishing. In addition, *S. trutta* was detected in the Tera River in autumn by eDNA metabarcoding. Therefore, the blanks were likely contaminated during the processing of the samples in the field or in the laboratory by the samples of the own sample locality and these species were not excluded from the L1 and L2 samples of each site.

Unassigned reads to our reference database of freshwater fishes (20.03% and 12.65% for M1 and M2, respectively) were aligned against the GenBank nucleotide database and, of these reads, only the 1.13% and the 1.54% were assigned to some species within Actinopterygii class (Fig 3). However, some of these reads had an alignment coverage with the matched sequence from GenBank lesser than 100% and/or the percentage of similarity lesser than 97%. After remove these unclear unassigned reads, we found that 92.5% and 96.8% of the unassigned reads for M1 and M2, respectively, were reliably assigned: the vast majority of reads were assigned to species within Mammal class (93% and 96% for M1 and M2, respectively), followed by Aves class (4% for both M1 and M2), Amphibia class (3% and 0.05% for M1 and M2, respectively) and Sauropsida class (0.04% for M1) and no unassigned reads were found for Actinopterigy class (Fig 3; S5 Table in S1 File). Some blank samples of both M1 (Arlanza, Pisuerga, Tera, Esla, Órbigo, Yeltes, Corneja, Voltoya rivers and Voltoya Reservoir) and M2 (Esla, Órbigo y Adaja), shown unassigned reads to our reference database, between 0–31.498

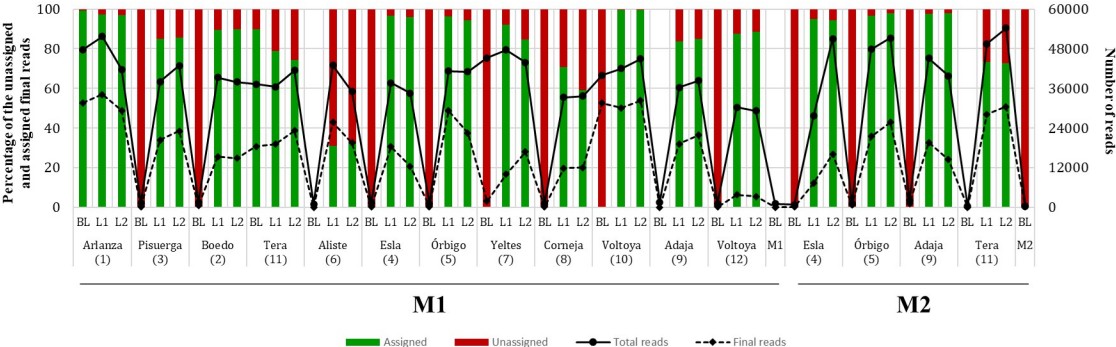

**Fig 2. Percentage of total reads assigned to some sequence from our taxonomy database.** The solid line indicates the number of total obtained reads and while the dashed line indicates the number of the total reads after the clean-up steps for each sampling point. The red and green bars indicate the percentage of the unassigned and assigned final reads to our taxonomy database, respectively. The samplings carried out in summer and autumn are indicated as M1 and M2, respectively. The name, number and type of sample (L1, L2 or BL) are also indicated with reference to those provided in Table 1.

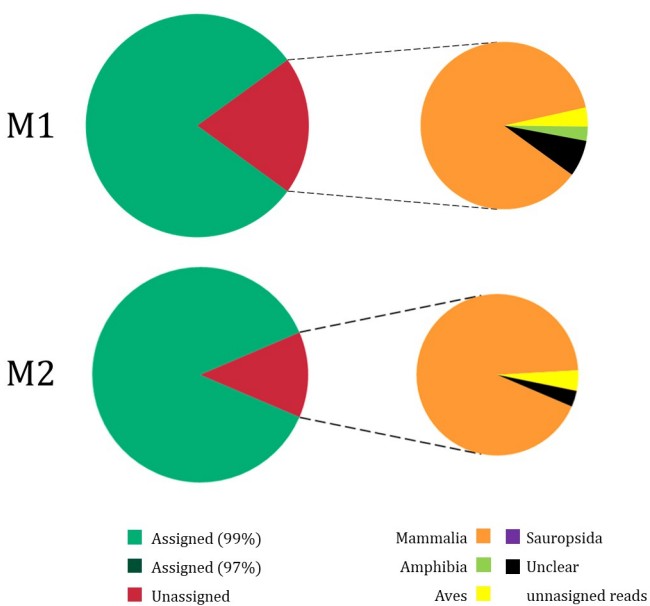

**Fig 3. Percentage of total assigned and unassigned reads to our database from the samplings carried out in either summer (M1, top) or autumn (M2, bottom).** The percentage of reads assigned with 99% identity (at the species level) and with 97% (at the genus level) are differentiated in green. In red unassigned reads to our reference database and aligned against GenBank database are shown and summarized by class. Reads with less than 100% GenBank matching sequence alignment coverage and/or less than 97% similarity are represented in black as unclear unassigned reads.

for M1 and between 0–1.064 for M2 (S5 Table in S1 File). After aligning them against a GenBank nucleotide database and removing the unclear assigned reads, we observed that for M1; 98,28% of unassigned reads corresponded to *Homo sapiens*, 1,56% to *Bos taurus*, and 0,16% to *Canis lupus*, while for M2; 78,53% of the reads corresponded to *H. sapiens* and 21,47% to *B. taurus*.

The reads assigned to fishes in our database with a percent identity of 99% or higher resulted in a taxonomic assignment of 1270 and 486 OTUs for M1 and M2, respectively; these values rose to 1525 and 533 OTUs, respectively, when percent identity of 97% was considered. The final matrix was constructed by combining the two matrices but maintaining the 99% assignment level when the OTU had been assigned in both cases (at the species level). As some species could not be differentiated at the 97% percent identity limit (Table 2; S3, S4 Tables in S1 File), the taxonomic assignment of the OTUs obtained at this percentage was done at the genus level, resulting in a taxonomic assignment of 1270 and 486 OTUs at the species level, and 255 and 47 OTUs at the genus level for M1 and M2, respectively. All genera assigned at 97% were present at 99% at the species level.

Nearly all of the native and non-native fishes species detected here by eDNA had been previously detected in the Duero Basin with some exceptions. As for the exceptions, *L. rifensis*, which is endemic to northern Morocco [36], was detected at a low proportion (x̄ = 0.03%) in the Arlanza River; *L. bocagei*, a related species, was detected at a higher proportion (x̄ = 3.6%) (S3, S4 Tables in S1 File). Given that *L. rifensis* does not occur in the Iberian Peninsula and its sequences are highly similar to those of *L. bocagei* (98.3%) we thought that these sequences could really be *L. bocagei* and it was excluded from the study. In addition, the introduction of freshwater fish species from North Africa in the Iberian Peninsula has not been previously reported. Also, reads assigned to genus *Leuciscus* (when analyzes were run at 97% percent

identity) was detected in Esla River in autumn. The only species of this genus present in the Iberian Peninsula is *L. aspius* [74], however, it is not known to occur in the Duero Basin. Low genetic differentiation for *12S* was observed between *L. aspius* and *A. alburnus* (97.7%), a species known to be present in that locality. In addition, reads assigned to the genus *Leuciscus* were only detected in autumn (M2) and when the analyzes were carried out with a percentage of similarity of 97%. Therefore, we considered that assigned reads to the *Leuciscus* genus correspond to the genus *Alburnus*, an invasive species in the Iberian Peninsula and extremely abundant in Duero Basin [75]. Therefore, we excluded *Leuciscus* genus from the study. In the Adaja River, *P. polylepis* was detected in both summer and autumn. This species has not been previously reported in the Duero Basin; however, its current distribution includes the Tajo Basin, which is adjacent to the southern part of the Duero Basin. We kept *P. polylepis* and discuss its putative presence in the discussion section.

It is also worth noting that, although marine fish species were not considered in this study, they were detected in the Adaja, Tera, Esla and Órbigo rivers in the summer sampling (M1), and in the Esla and Órbigo rivers in the autumn one (M2) (S3, S4 Tables in S1 File). These species included the European seabass (*Dicentrarchus labrax*), yellowfin sole (*Limanda aspera*), gilthead seabream (*Sparus aurata*), turbot (*Scophthalmus maxima*), European pilchard (*Sardina pilchardus*), albacore (*Thunnus alalunga*), Japanese jack mackerel (*Trachurus japonicus*), Atlantic horse mackerel (*Trachurus trauchurus*), European conger (*Conger conger*) and halibut (*Hippoglossus* sp.). These species are the most common in the Spanish diet. Therefore, excluding marine species, the *Leuciscus* genus and the species *L. rifensis*, a total of 22 species were detected across all sampling sites.

The number of species detected differed between the two sampling methods, regardless of whether only records of species detected by electrofishing of the present study were considered (Wilcoxon test, V = 78, $p$ = 0.002) or whether records of species detected by electrofishing of both present and past surveys were considered (Wilcoxon test, V = 59, $p$ = 0.02). A total of 22 freshwater fish species was detected by eDNA, and 19 by electrofishing (considering both present and past records). Considering all cases, that is, the detections for each species at each sampling point by electrofishing and eDNA metabarcoding, 27.95% were exclusively detected by eDNA, 5.37% by electrofishing and 66.66% by both methods (Table 3). However, all the cases in which the species were only detected by electrofishing were species that had already been found in the past (2001, 2009 and 2010) but were not detected in the electrofishing surveys carried out in the present study. In 17% of the cases, species were detected by eDNA metabarcoding that were also found in the past by electrofishing (2001, 2009 and 2010) but not at present. Detailed information on the percentage of relative abundance of reads obtained of L1 and L2, for each species and each sampling locality is provided in Fig 4. Also, in the 83.33% of the sampling points a higher number of species was detected by eDNA metabarcoding than by electrofishing survey (Fig 5A).

For the number of species in the 75% of the sampling sites, there were differences between L1 and L2. But these differences never exceeded in more than two species (Fig 5B). For the species that only were found in one of the liters, the percentage of relative abundance of its reads was always very low (<0.55%) except for two cases (1.29% and 2.48%) (Fig 4; S3, S4 Table in S1 File). There were no significant differences in the number of species detected between summer and autumn. However, the number of species detected for each season varied by 1, 2 or 4 species for Tera, Adaja and Esla, respectively (Fig 5C). Likewise, the mean relative abundance of the reads of all cases in which a species was detected in only one of the two seasons (M1 or M2) for the same locality ranged between 0.03% and 0.91%, except for two cases that presented higher values (2.6% and 3.70%) (Table 3).

**Table 3. Summary of the freshwater fish species detected using electrofishing (EF) and eDNA metabarcoding (ED).** The asterisk (*) in the EF section indicates that the species was detected in the samples from 2001, 2009 and 2010 by electrofishing but not in those of the present study. The pink color indicates that the species has only been detected by EF (5.38% of all cases). The blue color indicates that the species has only been detected by ED (27.5% of all cases). Purple indicates that the species has been detected by both methods (66.67% of all cases). Non-native species are marked with an asterisk.

Legend for cell fills below: P = purple (both methods), G = green/teal (ED only), K = pink (EF only), blank = not detected. An asterisk (*) marks EF detections from prior surveys.

| | Arlanza (1) EF | Arlanza (1) ED | Boedo (2) EF | Boedo (2) ED | Pisuerga (3) EF | Pisuerga (3) ED | Esla (4) EF | Esla (4) ED | Órbigo (5) EF | Órbigo (5) ED | Aliste (6) EF | Aliste (6) ED | Yeltes (7) EF | Yeltes (7) ED | Corneja (8) EF | Corneja (8) ED | Adaja (9) EF | Adaja (9) ED | Voltoya (10) EF | Voltoya (10) ED | Voltoya (11) EF | Voltoya (11) ED | Tera (12) EF | Tera (12) ED |
|---|---|---|---|---|---|---|---|---|---|---|---|---|---|---|---|---|---|---|---|---|---|---|---|---|
| *A. alburnus** | | | | | | | P | P | | | P | P | G | | | | | | | | | | | G |
| *A. arcasii (1)* | G | G | G | | G | G | * | P | | | G | G | | | P | P | P | P | P | P | P | P | P | P |
| *A. arcasii (2)* | | | | | | | | | | | | | | | P | P | | | | | | | | |
| *A. arcasii (3)* | | | | | | | * | P | G | G | | | | | | | | | | | | | | |
| *A. melas** | | | | | | | | | | | | | | | | | | | | | G | G | | |
| *A. salmantinum* | | | | | | | | | | | | | P | P | | | | | | | | | | |
| *B. barbatula* | | | | | | | * | P | | | | | | | | | | | | | | | | |
| *C. auratus** | | | | | | | | | | | | | | | | | G | G | | | *(K) | K | | |
| *C. calderoni* | P | P | * | P | | | | | | | P | | | | | | | | | | | | P | P |
| *C. carpio** | | | | | | | | | | | | | | | | | G | G | | | P | P | | |
| *C. paludica* | | | | | | | | | | | | | P | P | | | | | | | | | | |
| *E. lucius** | | | | | | | G | G | G | G | | | | | | | | | | | | | | |
| *G. holbrooki** | | | | | | | | | G | G | | | G | G | | | | | | | G | G | | |
| *G. lozanoi* | P | P | P | P | P | P | * | | * | P | G | | | | | | P | P | | | *(K) | K | P | P |
| *L. bocagei* | * | P | | | P | P | * | P | | | P | | P | P | | | | | | | P | | P | P |
| *L. gibbosus** | P | P | | | | | G | G | * | G | G | | | | | | | | | | | | | |
| *O. mykiss** | | | | | | | G | G | | | | | | | | | | | | | | | | |
| *P. bigerri** | P | P | | | P | P | * | | | | | | | | | | | | | | | | | |
| *P. duriense* | P | P | | | | | * | P | | | P | P | G | | | | | | | | P | | * | P |
| *P. polylepis* | | | | | | | | | | | | | | | | | P | G | | | | | | |
| *S. alburnoides* | | | | | | | | | | | | | | | P | P | * | K | | | | | | |
| *S. carolitertii* | * | P | * | K | | | G | G | | | P | P | P | P | | | | | | | P | | P | P |
| *S. trutta* | | | | | | | G | G | G | | | | | | | | | | G | G | G | G | * | P |
| *T. tinca** | | | | | | | * | P | * | P | | | * | P | | | | | G | G | *(K) | K | | |
| *Leuciscus* | | | | | | | G | G | | | | | | | | | | | | | | | | |

## Non-native species

Without considering the two species of undetermined origin, *T. tinca* and *G. lozanoi*, all sampling sites had a high number of non-native species except Boedo, Corneja and Voltoya rivers, which only presented native species. More non-native than native species were detected in the northwestern Duero Basin, specifically in the Esla, Órbigo and Aliste rivers (Fig 6). In the Esla, Órbigo, Yeltes, Adaja, Voltoya and Tera rivers were detected through eDNA metabarcoding non-native species that have not been previously recorded in these localities and they neither appeared in the present electrofishing survey (Table 3).

## Discussion

In recent years, advances in next generation sequencing (NGS) technologies have provided very useful tools for a variety of approaches in biodiversity conservation [76]. Specifically, eDNA metabarcoding has proven to be a powerful non-invasive tool for monitoring the biodiversity of freshwater fishes without the need for morphological identification [14, 77, 78].

We tested the efficiency of using eDNA metabarcoding with the *12S* marker to describe the composition of the freshwater fishes in the Duero Basin, the largest river basin in Spain. The

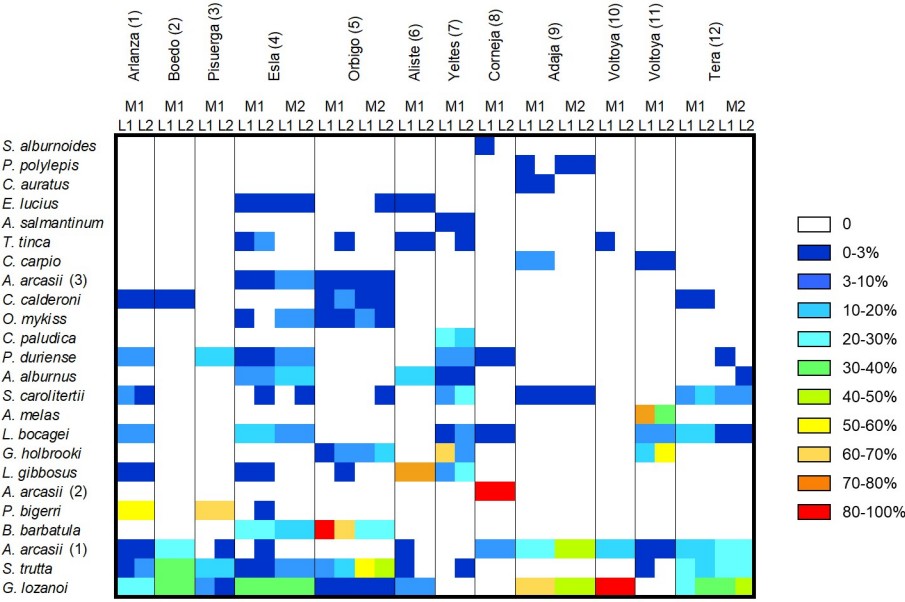

**Fig 4. Heatmap depicting the percentage of species detected for each liter (L1 and L2) of water sampled at each locality.** Samples of summer (M1) and fall (M2) are differentiated. The reads assigned at the genus level (at 97%) are not shown in the figure due to their low abundance but can be seen in S5 Table in S1 File.

sampling points of our survey covered a total representation of the ichthyofauna of the Duero Basin, with the exception of the species *C. vettonica*, distributed only in some small rivers of the southwestern Duero Basin that they have not sampled in the present study. Recent eDNA metabarcoding studies through *COI* or *18S* have been carried out in the Iberian Peninsula to

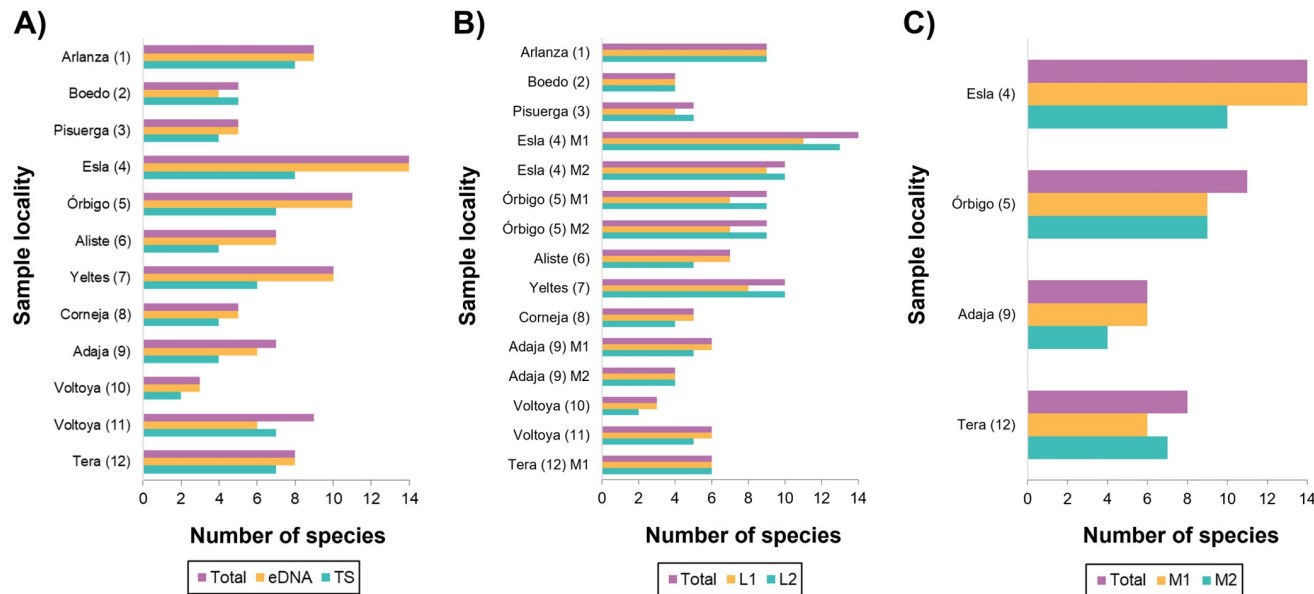

**Fig 5.** Comparison of the number of species detected for A) each detection method (eDNA and electrofishing for present and past results; TS), B) for each liter of water sampled at each locality (L1 and L2) and C) for each station (M1 and M2).

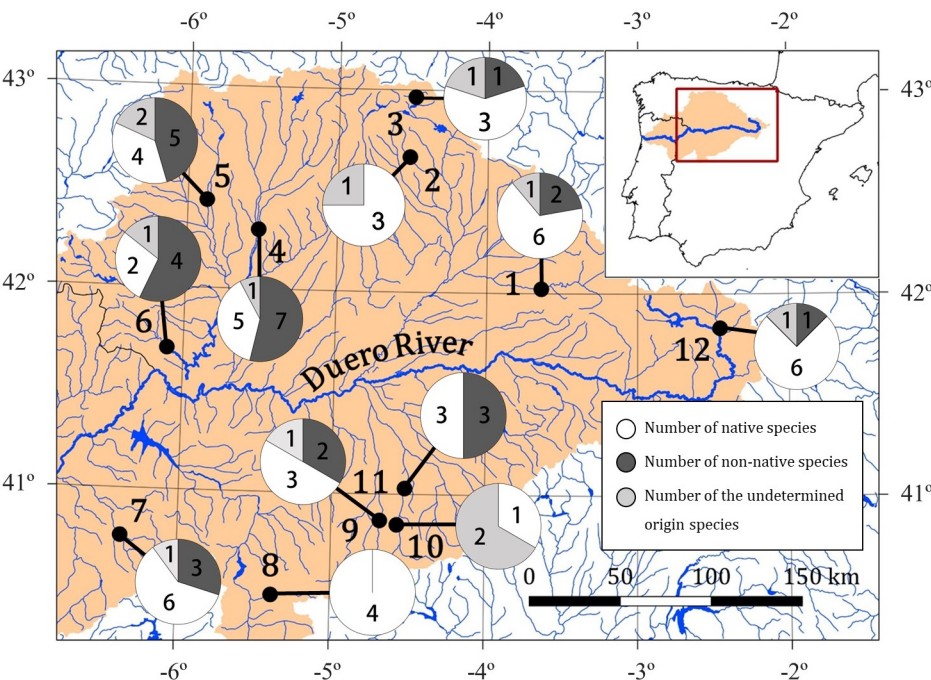

**Fig 6. Number of native (white) and non-native (dark grey) species present in the different sampling sites.** Species of undetermined origin (*G. lozanoi* and *T. tinca*) are indicated in light grey. Maps in this figure were made using the free software Qgis v.3.10.7. Inland water and administrative areas shapefiles from Spain were downloaded from data provided by DIVA-GIS project (https://www.diva-gis.org/gdata) and the Duero Basin shapefile was downloaded from data provided by the Ministerio para la Transición Ecológica y el Reto Demográfico (https://www.miteco.gob.es/es/cartografia-y-sig/ide/descargas/agua/cuencas-y-subcuencas.aspx).

study the intertidal meiofaunal communities or the total composition of macroinvertebrates and none of them was focused on the Duero Basin [79–81]. However, our study is the first to focus on determining the ichthyofaunal composition through the *12S* marker in the Iberian Peninsula.

## Traditional methods versus eDNA metabarcoding

The eDNA metabarcoding approaches have shown previously a higher capacity to detect the presence of freshwater fish species compared with traditional methods [15, 78, 82–84]. Consistent with these previous studies, we detected more species through eDNA metabarcoding than through traditional methods in the 83.33% of sample localities (see Fig 3).

Sampling through the eDNA metabarcoding is detecting DNA that is present in the river, but the individual is not detected, therefore, the results obtained on the presence data of a species through the eDNA metabarcoding must be interpreted with caution. Sometimes, DNA detected in one place may have originated from individuals in another due to indirect transport, giving rise false positives and a mischaracterization of the species composition of a given river [13, 85, 86]. In this context, false positives may arise mainly in two ways. First, the DNA detected could come from organisms located upstream. It has been shown that DNA can persist in water for several kilometers [26, 86, 87]. In our study, this phenomenon could explain the presence of the DNA of *A. arcasii* in the Arlanza or the Órbigo rivers. Although this species has never been recorded for either of these sampled sites by electrofishing, there are records of its presence in stretches upstream of these rivers at approximately 30 and 40 km for Órbigo

and Arlanza respectively [40, 42] which could explain the presence of DNA of *A. arcasii* in that stretch of the rivers. Second, false positives may be a consequence of DNA transported by other vectors such as bird excrements or human waste [82, 87, 88]. This explanation evidently accounts for the presence of the DNA of the marine species consumed as food in Spain commonly (i.e., *D. labrax*, *L. aspera*, *S. aurata*, *S. maxima*, *S. pilchardus*, *T. alalunga*, *T. japonicus*, *T. trachurus*, *C. conger* and *Hippoglossus* sp.) in the Esla, Órbigo, Adaja and Tera rivers, the four localities that presented recreational area. The presence of eDNA from fish species as a consequence of human remains has been previously detected in other eDNA metabarcoding surveys [89]. This explanation is also plausible, though less evident, for the presence of the DNA of the rainbow trout, *O. mykiss*, a non-native species in Spanish rivers that is also used as food, in the Esla and Órbigo rivers. The DNA of this species may be present due to both the presence of the species and the indirect transport of its DNA to these sampling sites. Notably, these rivers are near large urban centers, increasing the probability of finding eDNA derived from human remains.

Only 5.38% of the total of detected species were made exclusively by electrofishing. Even though a higher ratio of species is often detected by eDNA metabarcoding than by electrofishing, false negatives can occur for eDNA metabarcoding approach [13, 20]. False negatives may be associated with various factors such as the life cycle of a species, mainly associated with reproduction [90, 91], the abundance of a species [13, 15, 92, 93], river system dynamics [93], inappropriate markers [22, 82] or biases in amplification [20, 85]. In our case, the non-detection of taxa could be due to the absence of the species in the system since all the non-detections of eDNA metabarcoding had records from the past but were not captured in the present work by electrofishing. The majority of the detections made exclusively through electrofishing were for Juarros del Voltoya Reservoir (accounting for 60% of the total). This result was not unexpected as electrofishing was carried out in the vicinity of the reservoir but not in it. On the one hand, electrofishing sampling in the river showed a greater number of species, including *C. calderoni* and *G. lozanoi*, two benthic species. On the other hand, eDNA metabarcoding sampling in the reservoir showed fewer species, however *S. trutta* and *A. melas* were detected that had not been detected in the river by electrofishing, both used for fishing purposes. Likewise, the discrepancies in the composition of fishes between the reservoir and the river to which it flows have been sufficiently demonstrated for other places in the Iberian Peninsula [94, 95].

## Sample collection and rare species

The total volume of water collected for eDNA metabarcoding samples in previous eDNA metabarcoding surveys has varied from a few milliliters up to 50 liters [15, 96]. Recent surveys of freshwater fish detection through eDNA metabarcoding found that the volume of filtered water was a more limiting factor in detection of the number of species detected than the type of water body studied [97]. In addition, in a recent review of different methodologies used in eDNA surveys for freshwater fish species detection, it was noted that one to two liters of water was sufficient to obtain a representative sample of fish biodiversity [82, 97, 98]. In our case, we observed that 75% of the samples showed a difference in the number of species detected per liter for the same site, suggesting that one liter may be insufficient to fully capture the biodiversity of a site. We therefore recommend that a minimum sample volume of two liters of water. With respect to the samples taken at two different seasons (i.e., Esla, Órbigo, Adaja and Tera), we also observed a difference in the number of species detected in 75% of the samples regardless of the season sampled, although these differences were not significant.

Recent evidence suggests that eDNA concentration could be an indicator of abundance and/or biomass for fish stock assessments [27, 99]. Consistent with this, it has been shown that

less sampling effort is required to detect abundant species compared with less abundant ones [100, 101], and that species present at a low abundance have greater stochasticity in detection through eDNA metabarcoding [82]. All the species that were detected in either only one of the two liters (L1 or L2) or one of the two seasons (M1 or M2) were represented by a very low percentage of the relative reads (<3.7% of the relative reads). These species coincided with the ones that were not detected through electrofishing or that were detected in the samplings of the past (2001, 2009 and/or 2010) but not in those of the present study. There were only two exceptions; the first was for the detection of the species *S. alburnoides* in the Corneja, which was detected in only one of the two liters and by electrofishing in the present study, and the second was for the detection of the species *C. calderoni* in the Tera, which was detected in M1 by eDNA metabarcoding and electrofishing at present, but it was not detected for eDNA metabarcoding in M2. Therefore, it is possible that the stochasticity in the detection of some species through eDNA is likely explained by the low abundance of these species in the sampled rivers. However, the relation between species abundance and eDNA read number is influenced by multiple biotic and abiotic factors such as the taxon examined as well as their body size, distribution, reproduction, migration, water flow, temperature, and eDNA collection methods [99]. Therefore, to correlate the abundance of species and the number of reads obtained from the eDNA, an in-depth study of these rivers and the species inhabiting them would be necessary to identify the associated biotic and abiotic factors with the abundance of obtained reads from eDNA metabarcoding survey.

## The *12S* rRNA gene as a marker for eDNA metabarcoding studies

The *12S* rRNA gene fragment amplified with the MiFish primers [64] had sufficient taxonomic resolution to distinguish between the Iberian freshwater fish species studied in the present work, with a minimum similarity percentage of 99%. The percentage of similarity both within and between species was highly variable. Analysis of this marker revealed a higher interpopulation genetic variability than expected within *A. arcasii*. This species has a strong population genetic structure in the Duero Basin: three OTUs based on the *MT-CYB* marker had been previously identified for the species [37, 79], all three of which were recovered with the *12S* fragment in our analysis. These three groups presented a relatively low percentage of intraspecific genetic similarity (96%–97%). As the threshold percentage of similarity used in this study was 99% and 97%, the assignment of these species could have been confusing if the population structure of the species had not been known or considered. Therefore, we emphasize the importance of incorporating previous genetic studies for generating a reference database, to consider intraspecific variation and reduce taxonomic assignment errors. In the case of the Duero, we investigated the genetic variation within the populations of each species from previous molecular studies carried out with the *MT-CYB* [40]. Only the populations of *A. arcasii* showed a high genetic variation for *MT-CYB* [40], which agrees with the results obtained in this work for the *12S* marker. Finally, 1.13% and 1.54% of the reads for M1 and M2, respectively, were assigned to a species of Actinopterygii from GenBank but these reads had an alignment coverage with the matched sequence from GenBank lesser than 100% and/or the percentage of similarity lesser than 97%. One advantage of eDNA metabarcoding over conventional methods is the possibility of reanalyzing archived samples of metabarcoding results. Therefore, the development of a broader reference database in the future may reveal species that were not detected by previous studies [102, 103]. A surprising finding was the detection of the DNA of *P. polylepis* in the Adaja River, where *A. arcasii* was also detected by both eDNA metabarcoding and electrofishing. The native freshwater fishes of the Iberian Peninsula have undergone extensive speciation processed in the Iberian Peninsula after the formation of the

different basins [39]. In this way, many of the native species are restricted to one or a few basins, being the presence of two sister species in the same basin not common [39]. However, the hydrogemorphological changes of the Quaternary have explained different secondary contacts between basins and distribution patterns than usual [104]. Within Spain, the species *P. polylepis* is naturally distributed in the Tagus basin [45] and was introduced into the Jucar and Segura basins as a result of the passing of water of Tajo–Segura transfer [45]. However, currently, there are no records of the presence of *P. polylepis* in the Duero Basin. Stream captures (i.e., natural diversion of the channel of a river towards the neighboring river) between rivers in the southern part of the Duero Basin and those in the northern part of the Tagus Basin have been previously reported [105, 106]. These changes in the hydrogemorphology of rivers are sometimes associated with the exchange of fauna between two connected areas. In the Duero Basin only two exchanges of fish due to this secondary contact with Tajo Basin have been reported for the species *Squalius carolitertii* and *Cobitis calderoni* in two small rivers [32, 61]. The current hydromorphology of the Adaja River has been related to a possible stream capture and therefore could be associated with changes in the composition of the ichthyofauna [107]. In addition, from the species *P. duriense* was described, all the populations of the Duero Basin were considered as *P. duriense* (Coelho, 1985). However, a survey based on morphological characters found that the population from Adaja Sub-basin was morphologically more similar to the populations from Tajo Basin (currently considered as *P. polylepis*) than to the populations from Duero Basin [108]. Therefore, based on the above, we support that the species *P. polylepis* is present in the Adaja River.

In some rivers, *P. duriense* (in the Arlanza and Pisuerga rivers) and *A. arcasii* (in Corneja River) were detected by electrofishing, however, the eDNA analysis revealed the presence of the mtDNA of *A. arcasii* also in the Arlanza and Pisuerga rivers, and that of *P. duriense* in the Corneja River. Evidence of hybridization between some species of the genera *Achondrostoma* and *Pseudochondrostoma* has already been found for some populations within the Iberian Peninsula [35, 109]. Given that mitochondrial DNA is inherited maternally in most cases [110], it is not possible to distinguish hybrids from their maternal species when using eDNA approaches with a mitochondrial marker [13]. The same situation arises for the species *C. carpio* and *C. auratus*: they are well known to hybridize [111] and both their DNAs were found in the Adaja River. Therefore, for both cases it is likely both that the two species are coexisting in the rivers and that one of them has not been detected by electrofishing or that hybrids o hybrids and one of the parent species exists in these section of the rivers, which is undetectable for the eDNA metabarcoding approach through a mitochondrial marker.

## Conservation in Duero Basin

The presence of invasive species is one of the main threats to native freshwater fish fauna [51, 112]. Early detection of invasive species is, therefore, essential for the conservation of biological diversity [113–116]. However, this task can sometimes be hampered by the great effort required to detect a small number of individuals in large aquatic systems [117]. The high sensitivity of eDNA metabarcoding to detect species at a low abundance makes it an effective tool for locating potentially invasive rare aquatic or semi-aquatic species [11, 80, 117–120]. Through eDNA metabarcoding, we detected the first presence of different invasive non-native species at six of the sampling sites where they had not previously registered such as *G. hoolbroki*, *A. alburnus*, *A. melas*, *C. auratus*, *C.carpio*, *L. gibbosus* and *O. mykiss* (i.e., Esla, Órbigo, Yeltes, Adaja, Voltoya and Tera) [42, 63, 75]. This finding indicates the expansion of these species within the basin, and, in some of these cases (Esla, Órbigo and Aliste), the ratio of invasive to native species was higher. Therefore, the use of eDNA metabarcoding for monitoring the

freshwater fish composition within the Iberian Peninsula appears to be a useful tool not only for the early detection of invasive species, particularly those that are rare or elusive, but also for assessing the geographic distribution of the invasive species. Thus, an analysis and development of conservation and management strategies can be carried out considering these data, which will, in turn, influence the decision-making process and policy.

## Conclusion

Our survey has shown that eDNA metabarcoding is an adequate non-invasive monitoring method for the Iberian ichthyofauna present in the Duero Basin. We provided a genetic database for *12S* for all of the freshwater fish species present in the Duero Basin, which represent 30% of the species and 70% of the genera present in Spain. Some limitation has been detected for the eDNA metabarcoding method in the present study such as the inability to detect hybrids and the presence of false negatives and positives due to the eDNA metabarcoding technique being highly sensitive. However, the higher sensibility of this method compared to electrofishing makes it a powerful tool for different actions that can be carried out for the protection of biodiversity and aquatic ecosystems of the Duero Basin. On the one hand, this tool can be used for the early detection of invasive species, which are one of the main threats to the Iberian ichthyofauna [42] and therefore their early detection is essential for their conservation [118]. This tool can also be used for monitoring the effectiveness of eliminating invasive species from an ecosystem [121, 122]. On the other hand, non-invasive monitoring methods can be especially useful for threatened species whose viability may be compromised by direct capture, for example the endangered and Duero endemism *A. salmantinum* [8, 123]. This technique could also be useful for monitoring reintroduced species and to verify their success following their reintroduction [124, 125]. Because the costs of Next Generation Sequencing have been greatly reduced in recent years and eDNA metabarcoding techniques entail less sampling effort and lower cost for biodiversity monitoring [119, 126, 127], this could serve as a routine and complementary tool to traditional methods to control for the introduction and dispersal of both non-native and native species and to inform which management strategies are adequate for the conservation of the biodiversity of freshwater fishes in the Duero Basin.

## Supporting information

**S1 File.**
(DOCX)

## Acknowledgments

We want to thank P. Garzón, I. Doadrio Jr, J.L. González and G. González for collecting almost every sample from 2009 to 2010 and C. Marcos, J.C Velaco and G. González for their help in the project Life13 nat/es/000772. We greatly thank L. Alcaraz and M. Casal for laboratory and field assistance. We are grateful for the valuable suggestions and English editing performed by M. Modrell.

## Author Contributions

**Conceptualization:** Ignacio Doadrio.

**Data curation:** Andrea Corral-Lou.

**Formal analysis:** Andrea Corral-Lou.

**Investigation:** Andrea Corral-Lou, Ignacio Doadrio.

**Methodology:** Andrea Corral-Lou.

**Software:** Andrea Corral-Lou.

**Supervision:** Ignacio Doadrio.

**Writing – original draft:** Andrea Corral-Lou.

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
