## [Decision Letter · Decision Letter 0]

25 Aug 2022

PONE-D-22-09340Environmental DNA metabarcoding of water samples as a tool for monitoring Iberian freshwater fish compositionPLOS ONE

Dear Dr. Doadrio Villarejo,

Thank you for submitting your manuscript to PLOS ONE. After careful consideration, we feel that it has merit but does not fully meet PLOS ONE’s publication criteria as it currently stands. Therefore, we invite you to submit a revised version of the manuscript that addresses the points raised during the review process.

I got the recommendations and comments from an expert reviewer in the field. The reviewer agreed that the manuscript is technically sound and the data support the conclusions.

However, the lack of explanations, especially in the objective of this study and Results sections was suggested by the reviewer. Also, the reviewer pointed out the statistical tests on the results, I share their comments. Therefore, I can invite you to submit a revised version of the manuscript that addresses the points raised by the reviewers.

We look forward to receiving your revised manuscript.

Kind regards,

Hideyuki Doi

Academic Editor

PLOS ONE

Journal Requirements:

2. You indicated that IACUC animal ethics approval was not necessary for your study given the external permits granted by Dirección General de Medio Ambiente de la Junta de Castilla y León from Spain. We understand that the framework for ethical oversight requirements for studies of this type may differ depending on the setting and we would appreciate some further clarification regarding your research. Could you please provide further details on why your study is exempt from the need for approval and confirmation from your institutional animal research ethics committee (e.g., in the form of a letter or email correspondence) that ethics review was not necessary for this study? Please include a copy of the correspondence as an """"Other"""" file. Thank you very much for your attention to our requests

3. We note that you have stated that you will provide repository information for your data at acceptance. Should your manuscript be accepted for publication, we will hold it until you provide the relevant accession numbers or DOIs necessary to access your data. If you wish to make changes to your Data Availability statement, please describe these changes in your cover letter and we will update your Data Availability statement to reflect the information you provide

4. We note that Figure 1 & 6 in your submission contain [map/satellite] images which may be copyrighted. All PLOS content is published under the Creative Commons Attribution License (CC BY 4.0), which means that the manuscript, images, and Supporting Information files will be freely available online, and any third party is permitted to access, download, copy, distribute, and use these materials in any way, even commercially, with proper attribution. For these reasons, we cannot publish previously copyrighted maps or satellite images created using proprietary data, such as Google software (Google Maps, Street View, and Earth). For more information, see our copyright guidelines: http://journals.plos.org/plosone/s/licenses-and-copyright.

1. You may seek permission from the original copyright holder of Figure 1 & 6  to publish the content specifically under the CC BY 4.0 license.  

Additional Editor Comments (if provided):

I got the recommendations and comments from an expert reviewer in the field. The reviewer agreed that the manuscript is technically sound and the data support the conclusions.

However, the lack of explanations, especially in the objective of this study and Results sections was suggested by the reviewer. Also, the reviewer pointed out the statistical tests on the results, I share their comments. Therefore, I can invite you to submit a revised version of the manuscript that addresses the points raised by the reviewers.

Reviewers' comments:

Reviewer's Responses to Questions

**Comments to the Author**

1. Is the manuscript technically sound, and do the data support the conclusions?

Reviewer #1: Partly

2. Has the statistical analysis been performed appropriately and rigorously? 

Reviewer #1: No

3. Have the authors made all data underlying the findings in their manuscript fully available?

Reviewer #1: No

4. Is the manuscript presented in an intelligible fashion and written in standard English?

Reviewer #1: No

5. Review Comments to the Author

Reviewer #1: I enjoyed learning about a region and various species that I am not familiar with, and I think having a sound eDNA metabarcoding study across the Duero basin would be interesting and important. However, I think there are some major issues with this study that need to be addressed, and many portions of the manuscript that need to be reassessed and rewritten.

Here are some of the specific major issues that came up for me:

- The objective(s) should be very clear and specific and relate to what the authors did in the study. Currently the objective listed is extremely vague and there are a number of seemingly unrelated analyses and figures that follow. Based on other sections of the paper, it appears one of the main objectives was to compare eDNA metabarcoding to electrofishing samples. Be clear if this was the case. Based on paragraphs in the introduction on other mitochondrial markers, it seemed that another objective was to assess if 12S could be used to detect different fish species in this region. Was another objective to assess genetic differentiation or identify population-level differences for A. arcassii or other species based on known tissue samples? If it wasn’t, why was that included in the study and discussed at much length? There is also some discussion of invasive species throughout. Was that related to another objective? I think the authors should assess what the main points of this study are, identify specific objectives, and build their paper and figures with that in mind.

- It does not appear that the authors performed any statistical analyses to support their conclusions. There are several claims in the results and discussion (more species were detected with eDNA than traditional methods, species detected in summer were different than fall) but these must be backed up statistically. These results also need to tie into hypotheses related to specific objectives (see above comment).

- I find it difficult to trust the species assignments presented in the results. In the discussion, the authors acknowledge potential issues with assignments (for example, that the reads assigned to Leucisus might actually be A. alburnus because these two species are 97.7% similar). Was there any additional thought about other closely related species? What if a read had a 99% match to species A, and 98.7% match to species B, and a 98.5% match to species C… can we be totally confident that it is indeed species A? There is genetic variation among individuals within a species, and the variation between species will also differ for different genes. There are documented cases of some closely related species having a 99% or 100% match within the 12S gene. How can we be confident of our conclusions in such cases? The authors also acknowledge the high genetic variability for 12S within A. arcasii (~77%), but is there a potential for high interpopulation genetic variability within other species as well? What is the potential for misassignment of other species because of potential population variation, even if it has not yet been documented for other species in this region? I would like the authors to reflect and account for these potential problems, perhaps by assessing individual sequence matches and choosing to be more conservative and assign the read back to common genus or family if there is any ambiguity or uncertainty in species assignment.

-Much of the language throughout the manuscript is vague, confusing, unclear, or misleading. There are numerous grammatical errors, typos, informal language, and incorrect word choices throughout. I recognize the authors may use English as a second language, so I recommend using a language service or editor to iron out some of these issues. Even with fixing some of the minor language and grammar issues, however, many areas of the manuscript need to be explained, described, or clarified to much greater detail.

- If the fish species in this area lack data for the 12S gene, why was it used instead of the “markers of choice” brought up in the Introduction? If the point of this study was to show that the 12S gene can be used to detect native Iberian species the authors should be explicit about that and state that as a clear objective in the introduction. Otherwise, why weren’t the other genes used? Even if 12S had previously been shown to amplify local fish species, other metabarcoding studies have shown the utility of using more than one gene for more accurate results. Did the authors consider this? How could limitations of using only 12S affect results? This should probably be major part of the Discussion.

Specific comments organized by order of the manuscript are attached in a separate document.

6. PLOS authors have the option to publish the peer review history of their article (what does this mean?). If published, this will include your full peer review and any attached files.

Reviewer #1: No

---

## [Author Response · Author response to Decision Letter 0]

17 Feb 2023

Many thanks to the reviewer, his comments have served to significantly improve the manuscript. We then respond to each reviewer's suggestion.

Specific comments organized by order of the manuscript are listed below.

Abstract:

Line 26: What do the authors mean by primary freshwater fish? Does this mean the most common species? Unsure if “primary” is the correct word

Done. L96-97: We have added the definition of primary freshwater fishes: Primary freshwater fishes (i.e., whose ancestors entered inland waters much earlier, cannot survive in seawater and are thus strictly confined to fresh water) [29-31]

Line 33: Be clear that the population-level differences in 12S gene for A. arcasii was detected from tissue samples and not from eDNA. The way it is written now is misleading. 

Done. L34-35: We´ve added: and even showed population-level differences in the studied populations across the tissue samples for Achondrostoma arcasii.

Line 37: “Need to generate a local database” – the authors should be clear what kind of database this is referring to. I’m assuming a database for 12S gene sequences for known local fish species?

Done. L38-39: We´ve added: However, we highlight the need to generate a local genetic database for 12S gene …

Line 39: Final sentence in the abstract. Be explicit if the study showed eDNA can be an early detection method for invasive species, or if this is referring to eDNA studies in general

I’d like to see a brief mention in the abstract of how this study impacts knowledge/conservation/ management of Iberian peninsula fish populations specifically.

Done. L40-43: We´ve rewritten the sentence: Finally, our survey shows that the high detection sensitivity of eDNA metabarcoding and the non-invasiveness of this method allows it to act as a detection system for species of low abundance, such as early invasive species or species in population decline, two key aspects of conservation management of Spanish freshwater fishes.

Introduction:

Lines 51-59: Good points about the limitations of traditional sampling methods but I think the authors should also explicitly mention problems with gear bias and how that might lead to underestimating species composition and diversity (for example mesh size of nets, net location in water column or habitat)

Done. L54-57: We´ve added and rewritten: Sometimes, these traditional methods are hampered by the geomorphological characteristics of a river, net location in the water column or habitat, mesh size of nets, unfavorable weather and water conditions and study design (e.g., the need for legal permits to capture specimens or for experts to make the morphological identifications).

Lines 68-70: This last sentence could be expanded; what do the authors mean by each of these problems with eDNA? Explain and give examples of each. It’s probably also worth mentioning problems with contamination, false negatives and false positives and specifically how each of those could occur.

Done. L74-79: We have added: This technique also faces challenges associated with low-quality, low-quantity of DNA in the samples subject to other factors such as capture efficacy, extraction efficacy, the presence of inhibitors and contamination challenges which can be lead to presence of false positives or negatives [20,21]. Therefore, even if it has been demonstrated that this technique is highly sensitive to the detection of low DNA concentrations, appropriate critical experimental design considerations are essential.

Lines 71-85: This paragraph mostly seems like it would fit better in the discussion when mentioning any potential shortcomings of the study. The authors make a good point in mentioning other mitochondrial markers and limitations/benefits of each. I am also curious if the authors considered using more than one gene as other eDNA studies have done to reduce the potential shortcomings of using a single gene? I’d like to see some discussion about this, if not here then in the discussion.

Not done. We consider that this should come in the introduction since the selection of the marker that we use in the present study was made from a previous bibliographical study and the reasons for deciding the 12S fragment (which have shown that they are more efficient than the MT-CYB or COI) as well as its strengths and weaknesses, are exposed prior to the preparation of the study and are not part of the discussion of the study. However, the good taxonomic resolution of this fragment for Iberian freshwater fishes as well as the local reference database is a result of this study and is mentioned in the discussion section.

Line 82: “The reference database for 12S” – be clear about which database this refers to. GenBank?

Done. L92-95: We have added: Moreover, the reference database for 12S is not as extensive as those for MT-CYTB or COI markers in public repositories such as GenBank, widely used in phylogenetic and phylogeographic studies, making the taxonomic assignment of freshwater fishes more difficult [22].

Lines 86-92: If the fish species in this area lack data for the 12S gene why was it used instead of the “markers of choice”? If the point of this study was to show that the 12S gene can be used to detect native Iberian species the authors should be explicit about that and state that as a clear objective in the introduction. Otherwise, why weren’t the other genes used?

Done. L86-89: We have rephrased the sentenced and we think that now it is clearer: The strength of 12S markers for eDNA metabarcoding lies primarily in its more fish-specific amplification than other mitochondrial markers such as cytochrome c oxidase I (COI) and cytochrome b (MT-CYB) [22,24,25].

Done. L144-L146: We also have added this part in the main aims of the study: The main objective of this study is to assess eDNA metabarcoding as a tool for monitoring freshwater fish fauna from Duero Basin comparing with electrofishing results and to explore the use of 12S as a marker for this approach.

Line 87: “As with many other of these types of studies” – An example of vague wording. Be clear about what type of study this is referring to. eDNA studies? Phylogenetic studies?

Done. 98-102: We have rephrased it: All phylogenetic and phylogeographic studies in Iberian primary freshwater fishes have been conducted using the MT-CYTB and COI markers, while no surveys have been carried out with the 12S marker [36-41].

Lines 93-94: This paragraph should start with the description of the Iberian peninsula and then introduce the description of the Duero basin (start broad and zoom in).

Lines 94-97: Before introducing examples of threatened species, I would like to see some broader regional context and discussion of some of the problems within the basin that are leading to declines in certain fish populations. This is mentioned at the end of the paragraph but should be introduced sooner for readers who are not familiar with the region or the specific species mentioned earlier.

Lines 98- 100: The authors mention the distribution and population genetics of “several species” – be specific about which species these are and give examples. Give some context maybe earlier in the paragraph – approximately how many species exist in the peninsula or the basin? How many native vs introduced? How many is “several”? Are many species declining or just a few?

The tree suggesting done. L103-140: We have changed the entire paragraph and considered these three suggestions of the reviewer: Compared with other vertebrate groups in the Iberian Peninsula, the freshwater fishes in this area consist of a greater proportion of endemic species and represent one of the most endemic fish faunas in the Mediterranean region [42]. Specifically, of the 68 species present in the Iberian Peninsula, 55 are continental and 44 are endemic, and some of those that are not considered endemic have their range limited to a small portion of southern France, such as Phoxinus bigerri Rafinesque, 1820 and Barbatula hispanica (Lelek, 1987) [42-43]. The Duero Basin, with a surface area of 97,290 km², is the largest basin in the Iberian Peninsula (81% of it is in Spain, and 19% is in Portugal). The Duero Basin currently hosts 70% of the genera and 30% of Spain’s native freshwater fishes approximately, considering Gobio lozanoi Doadrio & Madeira, 2004 and Tinca tinca (Linnaeus, 1758), whose origin in the basin remains undetermined [42,44-45]. Of the native primary freshwater species found in the Duero Basin, 64% are included within one of the three most critical categories on the IUCN Red List (Vulnerable, Endangered and Critically Endangered; IUCN, 2020), including one of the most endangered Iberian endemic fish species, Achondrostoma salmantinum Doadrio & Elvira 2007 with its distribution area restricted to a few rivers in the southwestern Duero Basin. The main threats of these species are associated at anthropogenic factors, which in some cases in having led to an evident decline in populations of several of the species [42]. The main anthropogenic threat is the 18 large reservoirs and 3257 small dams that occur throughout the Duero Basin (https://www.miteco.gob.es; National River restoration strategy 2022-2030). This reservoirs or dams modify the dynamics of rivers, the habitat of native species, land use (93% of the reservoir water is used for irrigation) and are the main source of introduction of invasive species for recreational use [46-50]. Specifically, the Duero Basin hosts a large variety of non-native species, currently recorded a total of 15 non-native species (https://www.chduero.es/), being the presence of non-native species, a major threat to the sustainability of native species [51]. While the actual impact of invasive species on native species is often unknown, they can affect native wildlife through a variety of factors. Invasive species can compete for niches or resources, prey on native fauna, transfer pathogens, alter habitat, or cause genetic introgression, ultimately leading to loss of genetic diversity [52-56]. 

Duero Basin also harbors a great intraspecific genetic diversity for multiples native Iberian freshwater fish species, as revealed by multiple population genetics surveys [32,40,41]. On the one hand, for several species within this basin that are also distributed in other Iberian basins, such as Cobitis vettonica Doadrio & Perdices, 1997, Squalius alburnoides (Steindachner, 1866) and S. carolitertii (Doadrio, 1988), population genetics studies have revealed that Duero Basin populations behave as an independent evolutionary lineage [40,57,58]. On the other hand, a strong genetic structure within the populations distributed throughout the Duero Basin has been detected in other species such as C. calderoni Bacescu, 1962, Pseudochondrostoma duriense (Coelho, 1985) or Achondrostoma arcasii (Steindachner, 1866) [32,40,59]. For all these reasons, the conservation of freshwater fish populations in the Duero helps to protect not only the species richness of the Iberian Peninsula but also intraspecific genetic richness.



Lines 102- 104: “..64% are included in one of the three most critical categories..” – what does this mean exactly? What are the three categories? What is the “one” category this refers to? Be clear and specific.

Done. L113-115: We have added: Of the native primary freshwater species found in the Duero Basin, 64% are included within one of the three most critical categories on the IUCN Red List (Vulnerable, Endangered and Critically Endangered; IUCN, 2020)

Lines 106-107: “For several of the species”, again be specific about what “several” means and how that compares to other species, maybe give some examples. 

Done. We have changed the entire paragraph (see above)

Line 108: “High percentage” – percentage of what? Perhaps this should just refer to the number of different species in the basin rather a percentage of something. 

Done. L142-145: We have rewritten and added: Due to the large number of both native and non-native species inhabiting the Duero Basin in relation to Spain's entire freshwater fish fauna, the strong threat and the genetic richness of their native populations, accurate knowledge of the composition of this basin’s ichthyofauna is of critical importance for Spanish freshwater fish conservation.

Lines 108-110: This sentence could be broken up and clarified. Are the authors really concerned about the strong threat to exotic species? 

Done. We have added necessary information in the previous paragraphs.

Lines 110-114: The objective(s) should be very clear and specific and relate to what the authors did in the study. Based on other sections of the paper, it appears one of the objectives was to compare eDNA metabarcoding to electrofishing samples. Be clear if this was the case. Based on earlier paragraphs and discussion of other mitochondrial markers, it seemed that another objective was to assess if 12S could be used to detect different fish species in this region. Was another objective to assess genetic differentiation or identify population-level differences for A. arcassii based on known tissue samples? If it wasn’t, why was that included 1in the study?

Done. L145-143: We have rewritten and added: The main objective of this study is to assess eDNA metabarcoding as a tool for monitoring freshwater fish fauna from Duero Basin comparing with electrofishing results and to explore the use of 12S as a marker for this approach. For this purpose, we sequenced a fragment of 12S from tissue sample of all species inhabiting in the Duero Basin, tested the taxonomic resolution of this fragment and its ability to distinguish local variation and built a local 12S reference database for around of 30% of Spanish native freshwater fishes. Finally, we provide useful information on the presence of non-native species detected through eDNA, which is considered one of the main threats for Iberian native freshwater fishes.

Materials and Methods

Line 121: I’m not sure if “locality” is the best choice of word although I might be biased in my USA vocabulary. Something like “sampling site” or sampling location” seems more fitting for scientific studies

Done. We´ve changed locality by sampling site throughout the manuscript.

Line 123: What were the ichthyological samplings? What methods were used?

Done. L162-L164: We have added the sentence: The selection of the sampling sites was based on ichthyological electrofishing samplings made in 2001, 2009 and 2010 from previous projects attempting to obtain a representation of all the fishes that can be found in the Duero Basin [42,60].

Line 124: “The representative sample of all the fish..” Does this mean a representative sample of all the fish species present? Or some other metric? Be specific

Done. We have rewritten the sentence: See above.

Lines 126-127: How much time between eDNA sampling and electrofishing? This could be very important

Done. L164: We´ve added: The samples for eDNA metabarcoding were collected just prior to electrofishing in all sampling sites to avoid DNA contamination from the electrofishing gear between sampling localities.

Lines 128-130: The authors mention a reservoir but readers who are unfamiliar with this region don’t know what reservoir this is referring to. Perhaps some more background on the sample sites, or a description of in table 1 about what kind of system each site is (reservoir/lake/large river/small river/stream) would be helpful

Done. We´ve added the kind of system for each sample site in the Table 1.

Figure 1: This map is helpful, but coordinates should be labeled with units.

Done.

Table 1: Caption for the table should give descriptions of the columns. At this point the reader does not know what ID_M1 and ID_M2 mean. It would also be helpful to include some more site descriptions for people not familiar with this region, like I mentioned previously is it a lake/reservoir/river/stream? Is it in the headwaters or lower in the basin? What kind of land use is dominant in the area? Urban/farmland0/forestland? Highly impacted by humans or more “natural”? I would like to have some context to these sites. It would also be helpful to have the sampling dates of eDNA sample collection and traditional sampling at each site

Done. We´ve added: Table 1. Sampling sites include in the present study. For each sample site the following has been indicated: number on the map (Nº); Collection date of the samples in summer and autumn (Date M1 and Date M2 respectively); Name of the samples collected in summer and autumn (ID M1 and ID M2 respectively); Name of the river (River); The municipality in which it is located (Locality); The size of the aquatic system (Size); River section (Section); Land use type (Land use); Degree of impact of the river (Impact); Sub-basin to which it belongs (Sub-basin).

Lines 139-140: Please explain why only four sites were resampled in October and why those sites were chosen over others.

Done. L182-183: We´ve added: We also resampled four of the 12 sites in October 2020, which were associated with a recreational area and therefore a greater possibility of finding alien eDNA (M2; Table 1).

Lines 139-150: The authors should be more specific and detailed about the steps of collecting and filtering water samples and also build this paragraph in the order of the collection and filtering process so the reader can easily follow along. For example, filtering is currently mentioned before the description of water collection. More details are also necessary. How were water samples collected? From a boat or from the side of a river? Was the collector wearing gloves? Which equipment specifically was disinfected? What does filtering in situ mean? On a boat? On the side of a river? In a car? What was used to filter the water? Someone should be able to replicate exactly the sampling process based on the methods and right now a lot of the necessary detail isn’t included. 

Done. L183-199: We´ve added and rewritten some parts of the paragraph: All the used material was first disinfected with 20% bleach. The water samples were taken at two different points in the river, trying to cover all the heterogeneity of mesohabitats. The water was collected through a sterile bottle in the parts accessible to walk from the river with the use of gloves. A sample of the vertical water column was taken by submerging the bottle horizontally to the end and lifting it to the surface. In situ filtration was conducted on the riverbank to avoid cross-contamination and eDNA degradation. All water samples were filtered through Nalgene ™ Reusable Filter Holders with Receivers fitted with Supor®-200 Membrane Filters (Pall Corporation, Life Sciences, Ann Arbor, MI, USA) with a pore size of either 0.45 µm or 0.2 µm. ™ Reusable Filter Holders were connected to a vacuum pump which, in turn, was connected to a generator to allow the water to pass through the filters. For each locality, 1 L of distilled water was first filtered as a negative control (blank, BL) from the laboratory and then two 1-L samples of the river water (L1 and L2) were collected and filtered. Depending on the turbidity of the water, we had to use between one and three 0.45 µm-filters per liter. The filtered water samples were then re-filtered through a 0.2-µm filter to recover the maximum amount of eDNA present in the water. Each filter was stored directly in 1 ml of ATL buffer. Within 24 hours, 35 µl of proteinase K was added to each sample and it was digested for 15 hours at 56 ºC.

Lines 147-148: Why were filtered samples re-filtered through a 0.2 um filter?

Done. L196-198: We´ve added: The filtered water samples were then re-filtered through a 0.2-µm filter to recover the maximum amount of eDNA present in the water.

Lines 152-155: Be clear before describing this step that this is the exception to the protocol previously mentioned. 

Done. L201-202: We´ve added: The modification was the following: …

Lines 159-160: Who were the relevant authorities? 

Done. L210: We´ve added: (Environment service of council of Castilla y León)

Lines 158-166: Need more detail and clarification on how sample sites were selected and what kind of electrofishing was used. Where was the eDNA sampling site in relation to the electrofishing site at each location? The authors mention transects were sampled twice – how much time between the two surveys? What specific mesohabitats were surveyed? Wouldn’t there have been issues accessing some habitats while electrofishing? Again it would be useful to describe the whole process sequentially and in detail so the reader could easily replicate what the authors did.

Done. L208-210: We´ve added at the first of the paragraph: following the European regulations (EN ISO 14011:200. Water quality Sampling of fish with electrofishing)

L212-220: We´ve added a more detail information about electrofishing: Accessible River sections were selected to arrive by car and that it was possible to carry out electrofishing. At each sampling site, a river section of approximately 100 meters of distance just upstream from where the eDNA samples were taken, electrofishing sampling was carried out for 30 minutes approximately. The same transect was surveyed two following times separated by an hour. Electrofishing sampling was done in a zig zag covering all the accessible mesohabitats except in the Esla River (the only large river) where it was only possible to do electrofishing on the riverbank due to its great flow. For some specimens, a fin tissue sample was taken before they were returned to the river. Fin samples were preserved in 95% ethanol and stored at 4ºC until further processing. No individuals were sacrificed.

Lines 168-173: I think the authors need to be more specific on how these inventories were constructed. Did they create a list of all species observed from their electrofishing surveys and then add data from the mentioned previous projects? Was angler data only included in the Juarros de Voltoya site and not any others? The last sentence also mentions “data gathered in previous years”, the authors should be specific on what exactly this data is and what years specifically it was from.

Done. L222-225: We´ve added: An inventory of the ichthyofaunal composition was made at each sampling site. We created a list of all species observed from the current electrofishing survey and then we added the data from the list of all species recorded from previous projects led by the second author in 2001, 2009 and 2010 [42,60].

Line 175-176: Do the 48 samples include the 4 resampled sites? The wording is somewhat confusing

Done. L240: We´ve added: (including the 4 re-sampled sites).

Line 202: What is the unpublished data?

Done. L265: We´ve removed this affirmation because the paper has already published: The selection of species for the 12S reference database was based on previous knowledge of the ichthyofauna inhabiting the Duero Basin and genetic population structure studies [39-41].

Line 202-203: Which was the one species from Morocco?

Done. L266-268: We´ve added: The following 26 species (25 species present in Spain and Luciobarbus rifensis Doadrio, Casal-Lopez & Yahyaoui, 2015 from northern Morocco)

Line 215: Is 149 referring to the combined number of unique sequences across the 26 species previously listed? It would be helpful to know the range of number of sequences by species, for example did some species only have 1 sequence in GenBank while others had 40? 

Done. L281-283: We´ve added: Details of the number of specimens used for each species and the GenBank entrance number for each specimen are presented in S1 Table.

Lines 215-218: I’d like to see more detail about these additional sequences. Were the additional 19 species totally different from the previous 26? How much overlap was there? Again what was the range of the number of sequences per species? How old were the samples, how were they stored, and was there any concern for DNA degradation?

Done. L281-283: We´ve added: Details of the number of specimens used for each species and the GenBank entrance number for each specimen are presented in Table S1.

Line 231: How were sequences manually examined? Some more detail here would be useful.

Done. L295-299: We´ve added: All new electropherograms from the new sequences (78 sequences from tissue samples) were reviewed and cleaned one by one. Then all sequences (including GenBank sequences) were aligned using MAFFT [62], as implemented in Geneious 10.1.3 (http://www.geneious.com) [63] and collapsed into haplotypes.

Line 232-233: What was this already existing database? Is this different from the database the authors built? How were the sequences linked to this database? This description is lacking detail

Done. L298-300: We´ve added: Finally, to cover a wider spectrum of species, all our sequences were added to an already existing reference database which contains a trained reference sets that can be used to taxonomically assign fish 12S mitochondrial gene sequences [64,65].

Lines 235-238: The authors should give some context as to why this analysis was performed. Was an objective of the study to assess genetic differentiation across populations? What samples was this assessment based on? Known tissue samples or eDNA samples? If this assessment was based on tissue samples the authors should be clear and describe if they came from electrofishing samples or archived samples or other. This should either be explicitly included as an objective earlier in the manuscript or not included as it is not relevant to the eDNA assessment.

Done. L302-305: We´ve added: The taxonomic resolution inter-and intraspecific was examined through the genetic differentiation of the different sequences from both tissue and GenBank samples. This parameter was calculated to calibrate the parameters in the bioinformatics pipeline. 

We´ve also included it in the aims of the study in the introduction section

L303-308: We´ve also added: Genetic differentiation for 12S between and within species was based on the percentage of similarity calculated in Geneious 10.1.3. This parameter was calculated for the collapsed matrix of 46 haplotypes of the total of 227 sequences (149 from GenBank and 78 from the present study) for the 26 studied species.

Lines 252-257: The wording here is somewhat confusing. Were unassigned OTUs totally removed or were they mapped to GenBank to see if they matched to any species that might not have been in the authors’ database? The authors state they only kept reads that 100% aligned to known sequences from GenBank – what exactly does that percentage mean if they also set a 97% similarity threshold?

Done. L330-332: We´ve added and rewritten: In addition, to ensure the taxonomic assignment of the unassigned reads, we only kept reads whose cover of length of the alignment with a sequence from GenBank was 100% and had a percentage of similarity greater than 97%.

Did the authors consider doing any kind of statistical comparison in the number of species detected with each method and if it was significantly different? Or to test a difference in species between seasons for the 4 sites sampled twice? This is necessary to make any claims in differences in the Results and Discussion.

Done. We have performed the non-parametric Wilcoxon paired tests to compare differences in the number of species detected between sampling methods and between seasons. 

L231-237: We´ve added in the section of Inventory of ichthyofaunal composition: In addition, to test whether the number of detected species differed according to the sampling method used (eDNA metabarcoding and electrofishing) we used non-parametric Wilcoxon tests paired by all sample localities, since not all datasets were normally distributed. The comparison was carried out for both, considering the number of species detected by electrofishing including past and present data and considering only the present data. We also tested if there were differences in the number of species detected for summer and autumn through eDNA metabarcoding in Esla, Órbigo, Adaja and Tera.

We´ve added in the Results L453-458: The number of species detected differed between the two sampling methods, regardless of whether only records of species detected by electrofishing of the present study were considered (Wilcoxon test, V=78, p=0.002) or whether records of species detected by electrofishing of both present and past surveys were considered (Wilcoxon test, V=59, p=0.02).

We´ve added: L490: There were no significant differences in the number of species detected between summer and autumn.

Results

The order of the sections within the results should mirror the methods and put the most important parts of the study first 

Not done. The two sections of Material and Methods and that give rise to results are Genetic distance and Metabarcoding pipeline and it is in that order as shown in both sections

Line 260-262: It would be more useful to present this information in terms of ranges of different sequences per species and number of haplotypes per species. If this is referring to the results from known tissue samples the authors should be clear on that; the way it is presented now makes it sound like these results are referring to eDNA results which is confusing. Be clear on where the “new” sequences are coming from – electrofishing or archived samples or both?

Done. L335-338: We analyzed 227 sequences of 12S (149 from GenBank and 78 new sequences of tissue samples from DNA and Tissue Collection at the National Museum of Natural Sciences of Madrid) of 26 freshwater fish species (accession numbers of 78 new sequences: OP738998 - OP739075) (S1 Table).

Lines 262-270: See above comments on whether this analysis is relevant to the stated objectives and should even be included.

Not done. Knowing the taxonomic resolution for the 12S fragment is essential to perform a metabarcoding survey and thus know if through this marker we can know the population, species or the genus that inhabits the river. We´ve added in Material and methods L302-304: The taxonomic resolution inter-and intraspecific was examined through the genetic differentiation of the different sequences from both tissue and GenBank samples. This parameter was calculated to calibrate the parameters in the bioinformatics pipeline.

Table 2: Why were some groups of species assessed for interspecific similarity and not others? Is this analysis meaningful to the main objectives of the study?

Done. We´ve added extensive information in the caption of Table 2: Table 2. Summary of the percentage of similarity detected for the 12S gene within each species. Species pairs/groups showing an interspecific percentage of similarity >97% are also indicated. Non-native species are marked with an asterisk. Only the percentages of interspecific similarity between the species with values between 97-99% are shown. More detailed information on percentage of similarity between the different species is provided in S2 Table.

We have explored the percentages of similarity between the different species since in the Iberian Peninsula there are cryptic species that in recent years and thanks to the study with different molecular tools are giving rise to the description of new species. If we did not study interspecific (as occurred with A. arcasii) or interspecific variation, we could be losing information that may be important for conservation.

Lines 283-291: So when the authors blasted their unassigned reads they didn’t match to any fish specifically? Only to Actinopterygii as a broad group? Were there any matches to smaller groups of fishes? It seems odd that matches would jump from species level all the way up to Order I’d also like to see some discussion in this paragraph of the range of reads in the field blanks and lab blanks.

Done. We´ve clarify the sentence and we´ve added some information about unassigned reads in blanks. We also provide the Table S5 about unassigned reads in the supplementary material: L380-396: Unassigned reads to our reference database of freshwater fishes (20.03% and 12.65% for M1 and M2, respectively) were aligned against the GenBank nucleotide database and, of these reads, only the 1.13% and the 1.54% were assigned to some species within Actinopterygii class (Fig 3). However, some of these reads had an alignment coverage with the matched sequence from GenBank lesser than 100% and/or the percentage of similarity lesser than 97%. After remove these unclear unassigned reads, we found that 92.5% and 96.8% of the unassigned reads for M1 and M2, respectively, were reliably assigned: the vast majority of reads were assigned to species within Mammal class (93% and 96% for M1 and M2, respectively), followed by Aves class (4% for both M1 and M2), Amphibia class (3% and 0.05% for M1 and M2, respectively) and Sauropsida class (0.04% for M1) and no unassigned reads were found for Actinopterigy class (Fig 3; S5 Table).Some blank samples of both M1 (Arlanza, Pisuerga, Tera, Esla, Órbigo, Yeltes, Corneja, Voltoya rivers and Voltoya Reservoir) and M2 (Esla, Órbigo y Adaja), shown unassigned reads to our reference database, between 0-31.498 for M1 and between 0-1.064 for M2 (S5 Table). After aligning them against a GenBank nucleotide database and removing the unclear assigned reads, we observed that for M1; 98,28% of unassigned reads corresponded to Homo sapiens, 1,56% to Bos taurus, and 0,16% to Canis lupus, while for M2; 78,53% of the reads corresponded to H. sapiens and 21,47% to B. taurus.

Line 289: “The vast majority of reads.” Does this refer to non-fish reads? Or the reads after the authors excluded Actinopterygii? The language is somewhat confusing

Done. We´ve clarify this sentence. See in the paragraph above. 

Figure 2: Having the different y axes labelled would be helpful. I’m also not sure if using lines is the best way to present this data as lines typically show trends/changes, where here we are looking at one value (reads) per independent sample, therefore the lines are somewhat misleading. The caption should also define the components of the plot (site name, site number, BL, FL, L2, etc). Also does this plot show the reads for the sites that were sampled in October? Are reads from the two sampling times combined? The authors should be explicit if they are or otherwise. The sites sampled twice should also have some sort of symbol or asterisk to demonstrate which they are.

Done. We´ve added some information in the figure and we´ve added at the end of the Figure caption: The samplings carried out in summer and autumn are indicated as M1 and M2, respectively. Name, number and type of sample (L1, L2 or BL) are also indicated with reference to those provided in Table 1.

Not done. We find that the lines for each reading value make it easy to read for each sampling point and do not consider it to be overly confusing. 

Figure 3: It would be helpful to label M1 and M2 in the plot or just label them “summer” and “autumn”. I think the 99% and 97% label are somewhat confusing – either they should be called something else or defined clearly in the caption. The authors should clarify what they mean by “to the species level” in the caption. So is everything in green something that was assigned all the way down to species level or does it include identified genera or families too? Are we potentially missing out on data if reads were not assigned all the way down to species? It also appears that there is a typo in panel A – should the blue be labeled Actinopterygii? Why is this bold and underlined? Caption describing panel B should be clear that these are reads that were originally unassigned to the reference database and that these are the results after blasting to GenBank. Also think back to the main objectives and if this information on other orders is even meaningful or useful for the study

Done. We´ve changed the figure and the figure caption and now is clearer: Fig 3. Percentage of total assigned and unassigned reads to our database from the samplings carried out in either summer (M1, top) or autumn (M2, bottom). The percentage of reads assigned with 99% identity (at the species level) and with 97% (at the genus level) are differentiated in green. In red unassigned reads to our reference database and aligned against GenBank database are shown and summarized by class. Reads with less than 100% GenBank matching sequence alignment coverage and/or less than 97% similarity are represented in black as unclear unassigned reads.

Line 305: The authors should be explain this “matrix”; there was no previous mention of it.

Done. L326-328: We´ve added the information about this matrix in the Material and Methods section within Metabarcoding pipeline: We combined both matrices and keeping the OTUs assignments at 99% (at species level) for those OTUs that were assigned by both 99% and 97%. The OTUs that were only assigned to 97% were kept at the genera level.

Lines 307-310: It would be helpful to show any assignments to genus in the figures (specifically Figure 4) so the reader can see which genera were detected in which locations. Otherwise, we might be missing data in our interpretation of the results

Not done. The percentage of readings of the detected genera is always below 0.5 and, as explained in the text, the genera present always coincide with the genera of some species detected at the same point. We consider that adding this information could overload the image and does not provide additional information for the interpretation of the results. Anyway, that information is in the raw data table Table S3. We´ve added: The reads assigned at the genus level (at 97%) are not shown in the figure due to their low abundance, but can be seen in S5 Table.

Line 311: Are the 3 genera different from the 31 species or do they overlap?

Done. We have removed the sentence because it is confusing.

Lines 313-320: I would move this entire discussion of the marine species to a separate paragraph and keep the flow of the paragraph logical (introduce exceptions then immediately describe them and give examples)

Done. L443-452: We´ve moved this part to the end in another paragraph.

Lines 323-325: Do the authors think that the reads that matched to L. rifensis were actually from the closely related L. bocagei? If this is the case it should explicitly explained. Otherwise, is it totally impossible that L. rifensis could be in the basin? This might need to be explained for readers who aren’t familiar with this region and species. Or if the authors are unsure, couldn’t this assignment be to common genus?

Done. L423-430: We´ve added: As for the exceptions, L. rifensis, which is endemic to northern Morocco [36], was detected at a low proportion (x̄ = 0.03%) in the Arlanza River; L. bocagei, a related species, was detected at a higher proportion (x̄ = 3.6%) (S3 Table; S4 Table). Given that L. rifensis does not occur in the Iberian Peninsula and its sequences are highly similar to those of L. bocagei (98.3%) we thought that these sequences could really be L. bocagei and it was excluded from the study. In addition, the introduction of freshwater fish species from North Africa in the Iberian Peninsula has not been previously reported.

Line 325: Again, is it totally impossible that something in the Leuciscus genus could be in the basin just because it hadn’t been sampled before? How often did this genus appear in the samples? I am nervous that the authors may be losing meaningful results by discarding certain genera.

Done. L432-438: We´ve added more useful information for understanding: Also, reads assigned to genus Leuciscus (when analyzes were run at 97% percent identity) was detected in Esla River in autumn. The only species of this genus present in the Iberian Peninsula is L. aspius [71], however, it is not known to occur in the Duero Basin. Low genetic differentiation for 12S was observed between L. aspius and A. alburnus (97.7%), a species known to be present in that locality. In addition, reads assigned to the genus Leuciscus were only detected in autumn (M2) and when the analyzes were carried out with a percentage of similarity of 97%. Therefore, we considered that assigned reads to the Leuciscus genus correspond to the genus Alburnus, an invasive species in the Iberian Peninsula and extremely abundant in Duero Basin [72].

Lines 329-332: If the authors decided to keep P. polylepis that should be explicitly stated. 

Done. L441-442: We´ve added: We kept P. polylepis and discuss its putative presence in the discussion section.

Line 333: “A total of 22 species” – does this include genera that were not able to be classified down to species? If so, perhaps calling them “taxonomic units” or “taxa” would be better. If the authors are excluding any matches to genera/family, are they not losing potentially valuable data?

Done. L420-421: We have added in the Metabarcoding pipeline this sentence: All genera assigned at 97% were present at 99% at the species level.

Lines 334-342: I think it would be logical to move this paragraph earlier in the results when first discussing the number of reads and matches coming out of the metabarcoding pipeline. For the species that showed up in the field blanks – are they present in high numbers in those rivers i.e., does contamination from that species make sense? Is it possible they could have been contaminated in the lab too? (Perhaps this is more discussion material)

Done. L380-396: We´ve moved the entire paragraph.

Lines 343-345: This sentence is somewhat confusing and vague; each table and figure should be referred to clearly. 

Done. L464-466: We have rewritten the sentence and give more information: Detailed information on the percentage of relative abundance of reads obtained of L1 and L2, for each species and each sampling locality is provided in Fig 4.

Line 346: “Considering all cases” – what does this mean? The authors should avoid vague language and be explicit in what exactly they are referring to.

Done. L458-460: We´ve provided the “meaning” of cases: that is, the detections for each species at each sampling point by electrofishing and eDNA metabarcoding, 27.95% were exclusively detected by eDNA, 5.37% by electrofishing and 66.66% by both methods (Table 3).

Line 347-348: What do the authors mean by species detected by the traditional method were not in the present study? This language is confusing and makes it sound like there were no species detected in the authors’ electrofishing sampling

Done. L460-L464: We´ve rewritten: However, all the cases in which the species were only detected by electrofishing were species that had already been found in the past (2001, 2009 and 2010) but were not detected in the electrofishing surveys carried out in the present study. In 17% of the cases, species were detected by eDNA metabarcoding that were also found in the past by electrofishing (2001, 2009 and 2010) but not at present.

Figure 4: I think there is a better way to present these results. Perhaps have all the sites on one axis and all potential species on the other and create a kind of heatmap where shading of color represents number of reads (or percentage) for a specific species at a specific site. This would be easier for the reader to visualize species across sites and see which species are common, rarer, or might only be present in one or two locations. In the caption, define what percentage the authors are referring to – number of reads?

Done. We´ve changed the figure and now it looks like a heatmap: Fig 4. Heatmap depicting the percentage of species detected for each liter (L1 and L2) of water sampled at each locality. Samples of summer (M1) and fall (M2) are differentiated. The reads assigned at the genus level (at 97%) are not shown in the figure due to their low abundance but can be seen in S5 Table.

Figure 5: I think it would be best to just label each subplot rather than making the reader decipher the caption. Also the authors should be clear if the traditional methods results in the first panel include past samples or just the current electrofishing results. It might even be worth separating out past traditional sampling to the current study. I think the wording in the caption could also be more clear and list the panel letter before the description (e.g., “Comparison of the number of species detected for A) eDNA and traditional sampling (TS), B)….”)

Done. We´ve added different labels and in the Figure caption: Comparison of the number of species detected for A) each detection method (eDNA and electrofishing for present and past results; TS), B) for each liter of water sampled at each locality (L1 and L2) and C) for each station (M1 and M2).

Table 3: The caption refers to numbers in ED columns as mean relative abundance – does this mean percentage of reads in the sample aka the same values in Figure 4? The wording should stay consistent, or perhaps not be presented twice in a table and also in a figure. There is a lot for the reader to process and decipher in this table and I almost wonder if it should be broken up into more than one thing. If the percentage of reads is also presented in Figure 4 perhaps this table would be less overwhelming if it just included presence/absence of species in each separate method/sampling time, or perhaps separate out the comparison of past to current traditional sampling in a different table or figure. 

Done. We have changed the quantitative values for qualitative ones, indicating only if the species is present or not, and now the figure caption looks like: Table 3. Summary of the freshwater fish species detected using electrofishing (EF) and eDNA metabarcoding (ED). The asterisk (*) in the EF section indicates that the species was detected in the samples from 2001, 2009 and 2010 by electrofishing but not in those of the present study. The pink color indicates that the species has only been detected by EF (5.38% of all cases). The blue color indicates that the species has only been detected by ED (27.5% of all cases). Purple indicates that the species has been detected by both methods (66.67% of all cases). Non-native species are marked with an asterisk.

Lines 374-382: This entire paragraph is extremely difficult to follow, especially the second sentence. The authors should reflect on the main point they’re trying to make and explain it in a clear and simple manner. Instead of presenting percentages or variations in number of species in one or two liters, it would be much easier for the reader to comprehend if the authors just presented counts (e.g., the average number of species detected in each sample liter was 4.2 and ranged from 2 to 11). Also consider performing some kind of statistical comparison. 

Done. L488-493: We´ve rewritten the paragraph and now we think that is clearer: For the number of species in the 75% of the sampling sites, there were differences between L1 and L2. But these differences never exceeded in more than two species (Fig 5B). For the species that only were found in one of the liters, the percentage of relative abundance of its reads was always very low (<0.55%) except for two cases (1.29% and 2.48%) (Fig 4; S3 Table; S4 Table). There were no significant differences in the number of species detected between summer and autumn.

Line 378: What is a “station”? Is this the same thing as a sampling site or “locality? – wording should stay consistent throughout.

Done. L493: We´ve changed station by season.

Lines 383-389: I’m not sure how this section ties into the objectives. If a goal of the study was to compare native and introduced species detected with eDNA, that should be stated previously and described in the methods. 

Done. We have added as main objective to provide this information, which may be useful for future management and conservation plans, however, we do not treat it as a separate section in methods because the results for this are taken from the results obtained in the metabarcoding pipeline. We´ve add at the end of Introduction section: L151-153: Finally, we provide useful information on the presence of non-native species detected through eDNA, which is considered one of the main threats for Iberian native freshwater fishes.

We´ve added in the Inventory of ichthyofaunal composition section within Material y Methods section; L229-233: With the results obtained from this study, we provide useful information on the composition of the ichthyofauna, especially for non-native species.

Line 384: I am confused about the mention of the two species of undetermined origin. I do not remember seeing anything about this previously and I’m not sure what it means. Please explain this more fully either here or in the methods. 

Done. L110-112: We have clarified in the Introduction section: The Duero Basin currently hosts 70% of the genera and 30% of Spain’s native freshwater fishes approximately, considering Gobio lozanoi Doadrio & Madeira, 2004 and Tinca tinca (Linnaeus, 1758), whose origin in the basin remains undetermined [42,44,45].

Line 388-389: This sentence is very confusing and the authors should be specific in which rivers they’re referring to. Currently this makes it sound like no native species have been reported in some of the rivers which doesn’t make sense.

Done. L502-505: We´ve rewritten the sentence: In the Esla, Órbigo, Yeltes, Adaja, Voltoya and Tera rivers were detected through eDNA metabarcoding non-native species that have not been previously recorded in these localities and they neither appeared in the present electrofishing survey (Table 3).

Figure 6: Coordinates should be labeled with unit

Done.

Discussion (40) 

Sections of the discussion should reflect the objectives and main study goals

Lines 402-404: Does this mean C. vettonica was not included in the reference database or does it mean it was not detected in any of the eDNA samples? Be clear. If this result was not previously mentioned in the Results section (which I don’t remember it being stated) it should first be presented there. Remember, no new results should appear in the discussion.

Done. L517-519: We´ve rewritten the sentence and now is clearer: The sampling points of our survey covered a total representation of the ichthyofauna of the Duero Basin, with the exception of the species C. vettonica, distributed only in some small rivers of the southwestern Duero Basin that they have not sampled in the present study.

The species C. vettonica was included in the gene reference database just in case it appeared out of its distribution range. 

Lines 404-405: It would be useful to know what kinds of organisms these previous studies were on and how different the authors’ study is in comparison

Done. L520-523: We´ve added: Recent eDNA metabarcoding studies through COI or 18S have been carried out in the Iberian Peninsula to study the intertidal meiofaunal communities or the total composition of macroinvertebrates and none of them was focused on the Duero Basin [76-78].

Line 406: “a large number of..” is this referring to number of species in the reference database? In the eDNA results? Or both? One could argue most eDNA metabarcoding studies are assessing a “large number” of species.

Done. L523-524: We´ve removed and rewritten this part: However, our study is the first to focus on determining the ichthyofaunal composition through the 12S marker in the Iberian Peninsula.

Lines 409-410: Be clear that this refers generally to previous studies. Also are all of these studies on fish or aquatic organisms? 

Done. L526-527: We´ve rewritten this part: The eDNA metabarcoding approaches have shown previously a higher capacity to detect the presence of freshwater fish species compared with traditional methods [15,75,79-81].

Line 411: “with fewer than 100 species” why is that relevant? If this is important it should be mentioned in the Introduction but I would argue that this point isn’t important to the authors’ study.

Done. We´ve removed the sentence, not too much relevant.

Lines 411-413: It is difficult to make this claim without a statistical analysis of some sort. See previous comment in Methods

Done. He´ve done statistical analysis.

Line 413: Be clear what the 83.33% refers to. 

Done. L528-529: We´ve added: we detected more species through eDNA metabarcoding than through traditional methods in the 83.33% of sample localities (see Fig 3).

Lines 413-414: “However,..” this sentence could be worded more clearly. 

Done. L530-532: We´ve rewritten: Sampling through the eDNA metabarcoding is detecting DNA that is present in the river, but the individual is not detected, therefore, the results obtained on the presence data of a species through the eDNA metabarcoding must be interpreted with caution.

Lines 420-421: How far away are the locations upstream where the fish were recorded? Would this make sense with the eDNA results? Is it 1 km away? 100? Why wouldn’t the authors expect to detect this species in their sampling location? Remember to provide context for readers unfamiliar with these species and rivers.

Done. L538-542: We´ve added: Although this species has never been recorded for either of these sampled sites by electrofishing, there are records of its presence in stretches upstream of these rivers at approximately 30 and 40 km for Órbigo and Arlanza respectively [40,42] which could explain the presence of DNA of A. arcasii in that stretch of the rivers.

Lines 430-431: Very interesting about the marine species likely showing up from human consumption and waste. Were the sites that these species were detected downstream from urban areas or wastewater treatment plants? Are there other previous eDNA studies that have shown this phenomenon? It would be useful to compare and site them here.

Done. L547-548: We´ve added: The presence of eDNA from fish species as a consequence of human remains has been previously detected in other eDNA metabarcoding surveys [86].

Lines 431-434: Is there a citation to support this claim? In theory even if DNA is present in low concentrations, it should still be able to be amplified and detected, correct?

Done. We have removed the sentence.

Line 435: Be clear what this 5.38% percent refers to.

Done. L554-555. We´ve added: Only 5.38% of the total of detected species were made exclusively by electrofishing.

Line 437: “under this approach” – avoid vague terminology and be specific

Done. L556: We´ve added: for eDNA metabarcoding approach [13,20]

Lines 437-439: I’d like to see some more discussion and examples of these factors. For example, how would life cycle affect eDNA presence?

Done. L556-559: We´ve added: False negatives may be associated with various factors such as the life cycle of a species, mainly associated with reproduction [87,88], the abundance of a species [13,15,89,90], river system dynamics [90], inappropriate markers [22,79] or biases in amplification [20,82].

Lines 443-444: How far away from the reservoir did electrofishing take place? Could the discrepancy in species detected for Juarros del Voltoya also be explained by differences in fish communities of a lake habitat vs a river habitat? This might be good for the authors to discuss here

Done. L166-168: We´ve added in Sampling sites within Material and Methods section this sentence: In the sampling site Juarros de Voltoya (11; Table 1 and Fig 1), electrofishing was impractical within the reservoir itself, therefore we sampled the area just a few meters from the mouth of the reservoir in the Voltoya River. 

L564-570: We´ve also added in this part: On the one hand, electrofishing sampling in the river showed a greater number of species, including C. calderoni and G. lozanoi, two benthic species. On the other hand, eDNA metabarcoding sampling in the reservoir showed fewer species, however S. trutta and A. melas were detected that had not been detected in the river by electrofishing, both used for fishing purposes. Likewise, the discrepancies in the composition of fishes between the reservoir and the river to which it flows have been sufficiently demonstrated for other places in the Iberian Peninsula [91,92].

Lines 444-448: These points seem somewhat contradictory. If eDNA rapidly declines in lotic systems, wouldn’t eDNA sampled in lentic systems be more reliable? But this does not seem to be what the authors are saying. What do the authors mean by “spatial structure” of eDNA? They should be specific about what the degree of water mixing by seasons might do to eDNA and how that might affect their study.

Done. We have rewritten the sentence and we´ve removed this part (see above).

Lines 450-451: Is this referring to previous eDNA studies for fish? The authors should be specific

Done. L572-575: We´ve added: The total volume of water collected for eDNA metabarcoding samples in previous eDNA metabarcoding surveys has varied from a few milliliters up to 50 liters [15,93].

Line 451: The authors should be explicit about what “different optimal methodologies” means and what kinds of studies that refers to. Again, is this fish-specific?

Done. L575-577: We´ve added: in a recent review of different methodologies used in eDNA surveys for detection of fish species, it was noted that one to two liters of water were enough to obtain a representative sample of the biodiversity of fishes [94].

Line 455-456: I would also like to see some discussion on the potential problem of sampling at just one location in a large river system, and how specific sampling location might also affect results (i.e. collecting river samples from one location at a river bank, less than 1 m into the river in a river that might be 50 m across). As mentioned previously it would be useful to know the characteristics and water body size of the sampling locations. A single sample site might make sense in a small creek, but I would think in a large river you’d likely be missing a good amount of DNA present in the system, and I’d like the authors to reflect on this.

Done. L573-578: We´ve added: Recent surveys of freshwater fish detection through eDNA metabarcoding found that the volume of filtered water was a more limiting factor in detection of the number of species detected than the type of water body studied [94]. In addition, in a recent review of different methodologies used in eDNA surveys for freshwater fish species detection, it was noted that one to two liters of water was sufficient to obtain a representative sample of fish biodiversity [79,94,95].

Line 458-459: I’d like to see an explanation of how these fish species move and migrate throughout the year in this basin, or perhaps an example species from this study that would explain the difference in detection between summer and autumn. Otherwise, couldn’t the amplification of different species from one sampling time to the next be completely random? How do the authors know it’s not?

Done. We´ve removed this part of the discussion section because non-significant differences were found for number of species detected between M1 and M2.

Line 461: “reads” – be clear this refers to “read counts” or “quantity of reads”

Done. We´ve removed the first sentence. 

Lines 461, 463: This sentence is completely contradictory, perhaps change the wording

Done. We´ve removed the first sentence. 

Lines 467-468: These species weren’t detected in traditional sampling at all? How many species was this? How often were they only detected in eDNA vs also in traditional sampling? It would be useful to know the actual numbers, and if this isn’t presented in the Results it should be. 

Done. L591-597: We´ve added: These species coincided with the ones that were not detected through electrofishing or that were detected in the samplings of the past (2001, 2009 and/or 2010) but not in those of the present study. There were only two exceptions; the first was for the detection of the species S. alburnoides in the Corneja, which was detected in only one of the two liters and by electrofishing in the present study, and the second was for the detection of the species C. calderoni in the Tera, which was detected in M1 by eDNA metabarcoding and electrofishing at present, but it was not detected for eDNA metabarcoding in M2.

The detection of fish species through electrofishing survey was taken qualitatively, that is, presence/absence. Thus, we cannot provide the number of species detected. This information is given; “The percentage of relative reads of the cases where a species was detected in only one of the two liters of the same sample point was always lower than 0.55% except for two cases (1.29% and 2.48%) (S3 Table).”

Line 470-471: If the authors did observe a correlation between abundance and eDNA read count this should be listed as an objective and described in the Methods and the Results. Was this correlation observed for all species? Which ones? Also, some sort of statistical test on this relationship is necessary to make any conclusions.

Line 471: Explain some of these biotic and abiotic factors and give examples.

Lines 472-473: What kind of studies are the authors suggesting? They should be more specific. Water quality studies? Further fisheries sampling? Habitat surveys? Geomorphology?

Done all three. L599-605: We´ve rewritten the last part of the paragraph: However, the relation between species abundance and eDNA read number is influenced by multiple biotic and abiotic factors such as the taxon examined as well as their body size, distribution, reproduction, migration, water flow, temperature, and eDNA collection methods [96]. Therefore, to correlate the abundance of species and the number of reads obtained from the eDNA, an in-depth study of these rivers and the species inhabiting them would be necessary to identify the associated biotic and abiotic factors with the abundance of obtained reads from eDNA metabarcoding survey.

Lines 475-476: What about the instances that the 12S gene couldn’t classify down to species?

Not done. In all cases, the 12S gene was able to classify to species, which is what is discussed in this section. Moreover, what we discuss is that in some cases it even differed at the population level. Only when the percentage of similarity is lowered to 97 is it when it is not possible to distinguish between some very close species, but when the percentage of similarity is 99% it is possible to distinguish all the species studied. L609: We´ve added: with a minimum similarity percentage of 99%.

Line 482-484: The authors acknowledge the high variability within A. arcasii, but is there a potential for high interpopulation genetic variability within other species as well? What is the potential for misassignment of other species because of potential variation, even if it has not yet been documented for other species in this region? It is also well documented for some closely related species to have up to 100% similarity for a specific gene (including 12S), so how did the authors account for this possibility and ensure they were totally confident in species assignments? I’d like to see some discussion of this.

Done. We generated the reference database based on the current known genetic structure of the populations of the Duero basin. We have not seen variation between other fish populations in the Duero basin. If there was a very high variation, it is expected that the sequences would have been identified at a maximum of 97% and therefore they would be assigned at the genus level and not at the species level, so the assignment would not have been incorrect. Some species may have 100% similarity, which is why we carried out a preliminary study on the genetic differentiation of this gene for the populations of the Duero basin, considering previous phylogeographic studies carried out with other types of markers such as the MT-CYB. This explanation is given at the beginning of the paragraph. L619-623: We´ve added: In the case of the Duero, we investigated the genetic variation within the populations of each species from previous molecular studies carried out with the MT-CYB [40]. Only the populations of A. arcasii showed a high genetic variation for MT-CYB [40], which agrees with the results obtained in this work for the 12S marker.

Line 489: be specific about what archived samples this refers to; archived water samples? eDNA filters? Tissue samples? How long would these samples be good enough quality to reanalyze later?

Done. L626-627: We´ve added: One advantage of eDNA metabarcoding over conventional methods is the possibility of reanalyzing archived samples of metabarcoding results.

Lines 489-491: Using an additional gene for broader taxonomic coverage would also be beneficial

Not done. 12S has a sufficient taxonomic resolution, however there have been studies that show the efficacy of using various markers, but especially for groups that cannot be differentiated well. This is not the case in the Duero Basin with 12S.

Lines 493-500: I am confused how changes of two other rivers might have affected fish composition in the Adaja; please describe how these things connect and remember that readers likely aren’t familiar with all of these rivers so please provide some context.

Done. L629-652: We´ve rewritten and added some information: A surprising finding was the detection of the DNA of P. polylepis in the Adaja River, where A. arcasii was also detected by both eDNA metabarcoding and electrofishing. The native freshwater fishes of the Iberian Peninsula have undergone extensive speciation processed in the Iberian Peninsula after the formation of the different basins [39]. In this way, many of the native species are restricted to one or a few basins, being the presence of two sister species in the same basin not common [39]. However, the hydrogemorphological changes of the Quaternary have explained different secondary contacts between basins and distribution patterns than usual [101]. Within Spain, the species P. polylepis is naturally distributed in the Tagus basin [42] and was introduced into the Jucar and Segura basins as a result of the passing of water of Tajo–Segura transfer [42]. However, currently, there are no records of the presence of P. polylepis in the Duero Basin. Stream captures (i.e., natural diversion of the channel of a river towards the neighboring river) between rivers in the southern part of the Duero Basin and those in the northern part of the Tagus Basin have been previously reported [102,103]. These changes in the hydrogemorphology of rivers are sometimes associated with the exchange of fauna between two connected areas. In the Duero Basin only two exchanges of fish due to this secondary contact with Tajo Basin have been reported for the species Squalius carolitertii and Cobitis calderoni in two small rivers [32,58]. The current hydromorphology of the Adaja River has been related to a possible stream capture and therefore could be associated with changes in the composition of the ichthyofauna [105]. In addition, from the species P. duriense was described, all the populations of the Duero Basin were considered as P. duriense (Coelho, 1985). However, a survey based on morphological characters found that the population from Adaja Sub-basin was morphologically more similar to the populations from Tajo Basin (currently considered as P. polylepis) than to the populations from Duero Basin [106]. Therefore, based on the above, we support that the species P. polylepis is present in the Adaja River.

Line 495: Please define the Tajo-Segura transfer

Done. We have rewritten the sentence and now I more understandable (see above)

Line 496 and 500: Please define stream captures and stream piracy.

Done. Done. We have rewritten the sentence and now I more understandable (see above)

Line 498: Please describe what you mean by movement of ichthyofauna; temporary? Permanent? Migration?

Done. We have rewritten the sentence and now I more understandable (see above)

Line 500-501: Is this species known to be used for bait? Does that hypothesis make sense?

Done. We have removed this sentence because P. polylepis it is not used as live bait. 

Line 503: “In some rivers..” this should be the beginning of a new paragraph

Done.

Line 503-513: I agree that the potential for hybridization can confound eDNA results, but how can the authors be sure this is happening? I think it is a stretch to claim that “it is likely that hybridization zones exist for these species in these rivers”. It seems very possible that one of those species truly exists in that specific river and just wasn’t sampled by electrofishing. Or if those species are closely related enough to hybridize, how can the authors even be sure any of their species assignments are correct?

Done. The presence of hybrid populations can only be detected through nuclear markers since the single study for the mitochondrial marker only shows maternal inheritance and therefore non-recombinant loci of the species. We know that the species differ genetically enough by previous phylogeographic studies (Doadrio et al., 2021). The affirmation that hybrids have occurred is based on what usually happens for these species of the same genus in other places they can be found, they are shown as the presence of mitochondrial of both species and nuclearly. In any case, we mention both options.

Done. L662-665: We´ve added and rewritten: Therefore, for both cases it is likely both that the two species are coexisting in the rivers and that one of them has not been detected by electrofishing or that hybrids o hybrids and one of the parent species exists in these section of the rivers, which is undetectable for the eDNA metabarcoding approach through a mitochondrial marker.

Lines 518-520: The authors seem to be aware of the potential misassignment of species due to high similarity to other species, and based on this example of L. aspius and A. alburnus, the reader can’t help but wonder how much of these and other species detections can be trusted? I think the authors may need to go back and reassess some of their assignments, and if there are high percentage matches to multiple species, choose to be conservative and assign back to a common genus or family.

Done. We´ve remove this part in the discussion section because we remove this species from the results as is explaining in Results section: L430-438: Also, reads assigned to genus Leuciscus (when analyzes were run at 97% percent identity) was detected in Esla River in autumn. The only species of this genus present in the Iberian Peninsula is L. aspius [71], however, it is not known to occur in the Duero Basin. Low genetic differentiation for 12S was observed between L. aspius and A. alburnus (97.7%), a species known to be present in that locality. In addition, reads assigned to the genus Leuciscus were only detected in autumn (M2) and when the analyzes were carried out with a percentage of similarity of 97%. Therefore, we considered that assigned reads to the Leuciscus genus correspond to the genus Alburnus, an invasive species in the Iberian Peninsula and extremely abundant in Duero Basin [72]. Therefore, we excluded Leuciscus genus from the study.

Line 528-530: Please give examples of the invasive species detected. Are they all expanding? What are the authors using to compare this to and make this conclusion?

Done. We´ve added some cited about the presence of this non-native species in the Duero Basin: Through eDNA metabarcoding, we detected the first presence of different invasive non-native species at six of the sampling sites where they had not previously registered such as G. hoolbroki, A. alburnus, A. melas, C. auratus, C.carpio, L. gibbosus and O. mykiss (i.e., Esla, Órbigo, Yeltes, Adaja, Voltoya and Tera) [42,60,72].

I’m not sure if the concluding paragraph should solely discuss invasive species. Was assessing the presence of invasive species a main goal? The concluding paragraph should wrap up all the study objectives, summarize any limitations, and touch on specific management, conservation, or suggested future research.

Done. We´ve added a Conclusion part L685-706: Our survey has shown that eDNA metabarcoding is an adequate non-invasive monitoring method for the Iberian ichthyofauna present in the Duero Basin. We provided a genetic database for 12S for all of the freshwater fish species present in the Duero Basin, which represent 30% of the species and 70% of the genera present in Spain. Some limitation has been detected for the eDNA metabarcoding method in the present study such as the inability to detect hybrids and the presence of false negatives and positives due to the eDNA metabarcoding technique being highly sensitive. However, the higher sensibility of this method compared to electrofishing makes it a powerful tool for different actions that can be carried out for the protection of biodiversity and aquatic ecosystems of the Duero Basin. On the one hand, this tool can be used for the early detection of invasive species, which are one of the main threats to the Iberian ichthyofauna [42] and therefore their early detection is essential for their conservation [117]. This tool can also be used for monitoring the effectiveness of eliminating invasive species from an ecosystem [120,121]. On the other hand, non-invasive monitoring methods can be especially useful for threatened species whose viability may be compromised by direct capture, for example the endangered and Duero endemism A. salmantinum [8;122]. This technique could also be useful for monitoring reintroduced species and to verify their success following their reintroduction [123,124]. Because the costs of Next Generation Sequencing have been greatly reduced in recent years and eDNA metabarcoding techniques entail less sampling effort and lower cost for biodiversity monitoring [118,125,126], this could serve as a routine and complementary tool to traditional methods to control for the introduction and dispersal of both non-native and native species and to inform which management strategies are adequate for the conservation of the biodiversity of freshwater fishes in the Duero Basin.

General language issues:

- Choice of wording: I may be biased based on my country of origin (US) but I have never seen “autochthonous” and “allochthonous” used to refer to native vs introduced species. I find it distracting while reading but if those terms are commonly used and understood in other regions then ignore me. 

- Some choice of wording is unclear, vague, or misleading. Avoid words like “some” and “several” which appear often – descriptions should be as specific as possible. 

- Many sentences could be trimmed up and simplified; some are quite wordy and hard to follow due to confusing sentence structure or excess words that are unnecessary. For example line 158: “The traditional sampling approach of electrofishing was carried out..” could be “Electrofishing occurred..”

- This may be a personal preference, but the excessive use of passive voice is distracting

- Many phrases or word choices are informal or not suitable for scientific writing. Example line 511 “both their DNAs…” should be something like “the DNA of both species...”

We have revised the English of the entire document

---

## [Editor Report · Decision Letter 1]

2 Mar 2023

Environmental DNA metabarcoding of water samples as a tool for monitoring Iberian freshwater fish composition

PONE-D-22-09340R1

Dear Dr. Doadrio Villarejo,

We’re pleased to inform you that your manuscript has been judged scientifically suitable for publication and will be formally accepted for publication once it meets all outstanding technical requirements.

Kind regards,

Hideyuki Doi

Academic Editor

PLOS ONE

Additional Editor Comments (optional):

I carefully checked the revised manuscript as well as the response letter. I agree with the revisions and now can recommend publishing the paper.
---

## [Editor Report · Acceptance letter]

8 Mar 2023

PONE-D-22-09340R1 

Environmental DNA metabarcoding of water samples as a tool for monitoring Iberian freshwater fish composition 

Dear Dr. Doadrio Villarejo:

I'm pleased to inform you that your manuscript has been deemed suitable for publication in PLOS ONE. Congratulations! Your manuscript is now with our production department. 

Kind regards, 

on behalf of

Dr. Hideyuki Doi 

Academic Editor

PLOS ONE